# Quantification of epitope abundance reveals the effect of direct and cross-presentation on influenza CTL responses

Ting Wu[1,8], Jing Guan[1,8], Andreas Handel[2], David C. Tscharke [3], John Sidney[4], Alessandro Sette[4,5], Linda M. Wakim[6], Xavier Y.X. Sng[1], Paul G. Thomas [7], Nathan P. Croft [1], Anthony W. Purcell [1] & Nicole L. La Gruta[1,6]

The magnitude of T cell responses to infection is a function of the naïve T cell repertoire combined with the context and duration of antigen presentation. Using mass spectrometry, we identify and quantify 21 class 1 MHC-restricted influenza A virus (IAV)-peptides following either direct or cross-presentation. All these peptides, including seven novel epitopes, elicit T cell responses in infected C57BL/6 mice. Directly presented IAV epitopes maintain their relative abundance across distinct cell types and reveal a broad range of epitope abundances. In contrast, cross-presented epitopes are more uniform in abundance. We observe a clear disparity in the abundance of the two key immunodominant IAV antigens, wherein direct infection drives optimal nucleoprotein (NP)$_{366-374}$ presentation, while cross-presentation is optimal for acid polymerase (PA)$_{224-233}$ presentation. The study demonstrates how assessment of epitope abundance in both modes of antigen presentation is necessary to fully understand the immunogenicity and response magnitude to T cell epitopes.

[1] Monash Biomedicine Discovery Institute and Department of Biochemistry and Molecular Biology, Monash University, Clayton, VIC 3800, Australia. [2] Department of Epidemiology and Biostatistics, Health Informatics Institute and Center for the Ecology of Infectious Diseases, University of Georgia, Athens, GA 30602, USA. [3] John Curtin School of Medical Research, The Australian National University, Canberra, ACT 2601, Australia. [4] Division of Vaccine Discovery, La Jolla Institute for Allergy and Immunology, La Jolla, CA, 92037, USA. [5] Department of Medicine, University of California San Diego, La Jolla, CA, 92093, USA. [6] Department of Microbiology and Immunology, The Peter Doherty Institute for Infection and Immunity, University of Melbourne, Melbourne, VIC 3000, Australia. [7] Department of Immunology, St Jude Children's Research Hospital, Memphis, TN 38105, USA. [8] These authors contributed equally: Ting Wu, Jing Guan. Correspondence and requests for materials should be addressed to N.P.C. (email: nathan.croft@monash.edu) or to A.W.P. (email: anthony.purcell@monash.edu) or to N. Gruta. (email: nicole.la.gruta@monash.edu)

CTL responses are a critical determinant of protection against a number of diseases and there is a recognized need to develop vaccines that more effectively elicit CD8[+] T-cell immunity. The optimum design of any T-cell-based vaccine mandates a comprehensive understanding of the factors that govern peptide immunogenicity and the size of the immune response. It is likely that the context and differential abundance of epitopes presented by MHCI molecules plays a critical role in driving the reproducible CTL response hierarchies observed in MHC-matched individuals, especially during natural infection. Certainly, indirect analyses indicate antigen abundance plays a key role—alteration of epitope context or dose (or both) within a pathogen, for example, has the capacity to alter the size of the corresponding T-cell response[1,2]. Our appreciation of the impact of epitope abundance on antigenicity and CTL response magnitude has, until recently, been limited by a lack of sufficiently sensitive methodologies to accurately and specifically probe this parameter[3]. It is further complicated by the fact that MHCI-restricted epitopes may be presented by either directly infected cells or by professional antigen-presenting cells that take up infected material and exogenous viral antigens (cross-presentation). Each pathway may drive distinct epitope presentation characteristics for T-cell priming and expansion, yet the breadth and context of epitope abundance has not been studied systematically. Consequently, the question of how epitope abundance contributes to the immunogenicity and immunodominance of virus-derived CD8[+] T-cell epitopes remains poorly understood[4,5].

IAV infection of C57BL/6 (H-2[b]) mice elicits a CTL immunodominance hierarchy, with reproducibly large (immunodominant) responses directed towards the H-2D[b]-restricted epitopes $NP_{366–374}$ and $PA_{224–233}$, with all other detectable responses, including those directed toward $D^bPB1\text{-}F2_{62–70}$, $K^bPB1_{703–711}$, $K^bNS2_{114–121}$, and $K^bM1_{128–135}$ eliciting smaller (subdominant) responses[1,6]. Using an in vitro mass spectrometry-based strategy to identify and quantitate naturally processed H-2[b]-restricted peptides, we identify 21 IAV-derived peptides following both direct infection and cross-presentation. These include seven novel peptides, presented by H-2D[b] and H-2K[b]. All identified peptides are able to elicit CD8[+] T-cell responses following infection, indicating that epitope presentation following in vitro infection is representative of that during in vivo infection. MHCI-associated peptide abundance following direct infection is significantly correlated between dendritic cell (DC) and lung epithelial cells, reflecting conserved viral antigen expression and processing during infection across different cell types. Strikingly, the relative abundance of peptides presented following direct and cross-presentation highlights a remarkable disparity between the two immunodominant epitopes despite most peptides showing an overall correlation. Notably, the $PA_{224–233}$ peptide is much better presented via cross-presentation, while the $NP_{366–374}$ peptide is optimally presented following direct infection. The addition of quantitative MS data from directly infected cells and cross-presented viral antigen is used along with other variables (naïve T-cell precursor frequency, MHCI-binding affinity, protein abundance) to model the drivers of T-cell response magnitude to IAV. This analysis indicates significant contributions of both direct and cross-presentation, as well as peptide affinity for MHCI, in establishing the IAV-specific CD8[+] T-cell immunodominance hierarchy.

## Results

### Identification and quantitation of MHCI-bound IAV peptides.
Although a number of CD8[+] T-cell epitopes have been identified in the B6 model of IAV infection[7–9] (Table 1), their identification

has been predominantly achieved through epitope prediction and screening of T-cell responses. To define a more complete spectrum of MHCI-bound IAV-derived peptides presented following infection, we employed a conventional mass spectrometry approach (LC-MS/MS) to detect H-2D[b]- and H-2K[b]-bound viral peptides following in vitro infection of DC2.4 cells[10]. Infected DC2.4 cells were harvested and lysed at 8 h post infection (hpi), peptide-MHCI complexes (pMHCI) were isolated, and peptides analyzed by LC-MS (Supplementary Data 1). A total of 21 IAV-derived MHCI-bound peptides were identified. These included the $NP_{366–374}$ [D[b]], $PA_{244–233}$ [D[b]], $NS2_{114–121}$ [K[b]], $PB1_{703–711}$ [K[b]] and $PB1\text{-}F2_{62–70}$ [D[b]] peptides to which T-cell responses have been extensively characterized[1,6], and seven novel peptides that had not been previously reported (Table 1). Of these novel peptides, three were derived from the HA protein, two from the M1 protein, and one each from NA and NS2. $HA_{41–49}$ (VTVTHSVNL) was detected in both K[b] and D[b] eluates, indicating that this peptide is a promiscuous binder, whilst $NS2_{8–16}$ was the only novel K[b]-binder (Table 1). Although mouse MHCI K[b] and D[b] molecules typically bind peptides of 8–10 aa in length, three of the seven newly identified peptides were 11 aa long. The length of the newly identified peptides likely contributed to their obscurity to date since previous publication of a widely tested potential repertoire of IAV-derived epitopes generated via a matrix-based algorithm, assumed lengths of 8 aa for K[b] binders and 9–10 aa for D[b] binders[9]. In summary, this screening process culminated in the identification of 21 IAV peptides presented by either H-2K[b] or H-2D[b] (or both for $HA_{41–49}$), including all of the well-characterized epitopes known to elicit CD8[+] T-cell responses.

### Quantitation of directly presented IAV peptides.
Having established the detectable repertoire of IAV peptides directly presented by DC2.4 cells at 8 hpi, we next quantitated the abundance of each peptide. The IAV peptides isolated from H-2D[b] and K[b] molecules were quantitated using an LC-MRM (liquid chromatography-multiple reaction monitoring) approach as previously reported for other viral epitopes[4] (Supplementary Data 2). Stable isotope-labeled viral peptides were synthesized and LC-MRM parameters individually optimized for each peptide to provide a complete suite of internal quantitative standards (see Table 1, Supplementary Tables 1–3, and Supplementary Figs. 1 and 2). The abundance of the 21 IAV-derived peptides spanned three orders of magnitude, ranging from 1–2 copies/cell of $PB1_{653–660}$ to an average of 3871 copies/cell of $NP_{366–374}$ (Fig. 1a). Of the well-characterized CD8[+] T-cell epitopes, the immunodominant $NP_{366–374}$ and subdominant $NS2_{114–121}$ peptides were the most abundant, being present at an average of 3871 and 2464 copies/cell, respectively, while the subdominant epitopes $PB1\text{-}F2_{62–70}$ and $PB1_{703–711}$ were substantially lower at 684 and 294 copies/cell, respectively. One of the least abundantly presented species was the immunodominant epitope $PA_{224–233}$, at only 7 copies/cell. Thus, the abundance of peptides presented following direct infection of the DC2.4 cells did not predict the CTL immunodominance hierarchy.

Given that productive IAV infection is restricted to respiratory epithelial cells, which are also the targets of the IAV-specific CTL response, we next investigated the relative abundance of IAV-derived peptides presented on H-2D[b] and K[b] molecules expressed on the surface of an infected lung epithelial cell line (LET1 cells)[11] (Supplementary Data 2). Although the infection efficiency of LET1 cells was similar to DC2.4 cells (~80–85%) (Supplementary Fig. 3a), the expression of surface H-2K[b] and D[b] complexes was lower in LET1 cells (Supplementary Fig. 3b), resulting in an overall reduction in the yield of peptides/cell (Fig. 1b). However,

## Table 1 Peptides identified from IAV-infected cells by LC-MS/MS

| Peptide | Sequence | Length (aa) | Allele | Citation |
|---------|----------|-------------|--------|----------|
| NP₃₆₋₄₃ | IGRFYIQM | 8 | $K^b$ | 9 |
| NP₅₅₋₆₃ | RLIQNSLTI | 9 | $D^b$ | 70 |
| NP₃₆₆₋₃₇₄ | ASNENMETM | 9 | $D^b$ | 71 |
| PA₂₂₄₋₂₃₃ | SSLENFRAYV | 10 | $D^b$ | 72 |
| PB1₆₅₃₋₆₆₀ | KNMEYDAV | 8 | $K^b$ | 9 |
| PB1₇₀₃₋₇₁₁ | SSYRRPVGI | 9 | $K^b$ | 73 |
| PB1-F2₆₂₋₇₀ | LSLRNPILV | 9 | $D^b$ | 74 |
| PB2₂₂₇₋₂₃₄ | VYIEVLHL | 8 | $K^b$ | 7 |
| **NS2₈₋₁₆** | **SFQDILLRM** | **9** | **$K^b$** | |
| NS2₁₁₄₋₁₂₁ | RTFSFQLI | 8 | $K^b$ | 7 |
| M1₁₂₈₋₁₃₅ | MGLIYNRM | 8 | $K^b$ | 7 |
| **M1₂₀₇₋₂₁₆** | **SQARQMVQAM** | **10** | **$D^b$** | |
| **M1₂₂₇₋₂₃₇** | **AGLKNDLLENL** | **11** | **$D^b$** | |
| **HA₄₁₋₄₉** | **VTVTHSVNL** | **9** | **$D^b$** | |
| **HA₄₁₋₄₉** | **VTVTHSVNL** | **9** | **$K^b$** | |
| HA₃₀₄₋₃₁₁ | SSLPYQNI | 8 | $K^b$ | 8 |
| **HA₃₀₈₋₃₁₆** | **YQNIHPVTI** | **9** | **$D^b$** | |
| **HA₃₈₉₋₃₉₉** | **NGITNKVNTVI** | **11** | **$D^b$** | |
| HA₄₀₂₋₄₀₉ | MNIQFTAV | 8 | $K^b$ | 9 |
| NA₁₈₁₋₁₉₀ | SGPDNGAVAV | 10 | $D^b$ | 9 |
| **NA₁₈₁₋₁₉₁** | **SGPDNGAVAVL** | **11** | **$D^b$** | |

Sequences in bold represent novel epitopes identified in this study
*IAV* influenza A virus

there was a significant correlation (RCC = 0.6327, p = 0.0021) between the relative mean abundance of each of the peptides quantified from both cell lines (Fig. 1c). To determine how the relative abundance of peptides presented in the context of H-2$K^b$ or $D^b$ molecules correlated with the relative amount of source protein present at 8 hpi, we determined relative protein abundance using label-free quantitation (LFQ) through analysis in Maxquant[12]. A Spearman's rank correlation test revealed no correlation between relative source protein abundance and peptide abundance, which was exemplified by the fact that multiple peptides derived from the same source protein (e.g. NP₃₆₆₋₃₇₄, NP₃₆₋₄₃, and NP₅₅₋₆₃) showed remarkably different abundance profiles (Fig. 1d). Collectively, these data indicate that the relative abundance of IAV-derived peptides presented by MHCI following direct infection was conserved irrespective of the cell type infected. Moreover, the abundance of the source protein at 8 hpi was a poor predictor of the relative level of MHCI presentation.

**Kinetics of IAV peptide presentation after direct infection.** Unlike members of the orthopoxvirus or herpesvirus genera, in which discrete early, intermediate and late waves of transcription are readily defined, IAV has a segmented RNA genome that is simultaneously transcribed and is therefore unlikely to show large variation in protein expression kinetics. To investigate whether IAV-derived peptide presentation was also temporally uniform, cell lysates were harvested from IAV-infected DC2.4 cells at various timepoints post infection and viral epitope abundance determined by LC-MRM (Fig. 2a and Supplementary Fig. 4 and Supplementary Data 2). Six of the 21 epitopes were detectable by 30 min after infection, including all three of the M1 peptides (M1₁₂₈₋₁₃₅, M1₂₀₇₋₂₁₆, M1₂₂₇₋₂₃₇), both of the NA peptides (NA₁₈₁₋₁₉₀, NA₁₈₁₋₁₉₁), and one of the HA peptides (HA₄₁₋₄₉), indicating some preference for structural protein-derived peptides at this early timepoint. The majority of the remaining epitopes could be detected by 2.5 hpi, and all epitopes were detectable by 4.5 hpi. The subsequent kinetics of peptide presentation fell broadly into three categories: seven peptides whose presentation

peaked at around 6.5 hpi and gradually diminished thereafter (Fig. 2a, yellow), 12 peptides whose presentation peaked at around 9.5 hpi and subsequently diminished (Fig. 2a, red), and two peptides whose presentation was continuing to increase at 12.5 hpi (Fig. 2a, blue). Neither of the immunodominant NP₃₆₆₋₃₇₄ or PA₂₂₄₋₂₃₃ epitopes showed a more rapid presentation kinetic than subdominant determinants, nor was there an association between duration of presentation and immunodominance, with NP₃₆₆₋₃₇₄ presentation declining at 12.5h (Fig. 2a, dashed lines).

Given the suggestion that peptide presentation may be more closely associated with the rate of protein translation rather than steady-state protein amounts[4,13], we determined the relationship between the kinetics of protein expression and peptide presentation. Antigen expression was detectable either prior to, or coincident with, the detection of presented peptide. Intriguingly, peak presentation of the majority of the peptides (15/21) preceded the peak of protein abundance, while the peak of the seven remaining peptides was coincident with peak antigen expression (Fig. 2b). These observations corroborate studies suggesting proteasomal degradation of newly synthesized proteins is a prominent source of MHCI peptides[14]. Again, as for absolute abundance at 8 hpi, the kinetics of peptide presentation were not dictated by the protein source as peptides derived from the same protein exhibited distinct kinetics. Thus, the majority of peptides were optimally presented between 6.5 and 9.5 hpi and the kinetics of epitope presentation occurred largely independently of antigen expression kinetics.

**Identification and quantitation of cross-presented peptides.** Cross-presentation has been implicated in the generation of CTL immunity during IAV infection[15]. Having established the abundance hierarchy of each of the IAV peptides following direct infection of both a DC and a lung epithelial cell line, we next sought to determine whether a similar or distinct peptide hierarchy was observed following cross-presentation.

To assess cross-presentation of IAV-derived peptides (Fig. 3a), MHCI $D^{b-}$ $K^{b-}$ donor cells (human alveolar epithelial cell line, A549) were infected with IAV, irradiated and washed, and then incubated with a CpG-activated, cross-presenting MHCI $D^{b+}$ $K^{b+}$ Mutu DC line[16]. Cross-presentation of IAV-derived peptides was confirmed using IAV engineered to express the ovalbumin-derived SIINFEKL epitope[17]. Cross-presentation of the $K^b$-SIINFEKL epitope was detected both via cell surface staining of Mutu DCs with an antibody specific for the $K^b$-SIINFEKL complex (Supplementary Fig. 5a), as well as by division of OT-I cells following 48 h co-culture with cross-presenting Mutu DCs (Supplementary Fig. 5b).

Having established the in vitro cross-presentation system, the abundance of IAV-derived peptides presented via cross-presentation by Mutu DCs was determined by LC-MRM (Supplementary Data 2). Compared to directly presented peptides (Fig. 1a), the abundance of cross-presented peptides was substantially reduced (the most abundantly cross-presented peptide, PB1₇₀₃₋₇₁₁, was presented at an average of 39 copies/cell, compared to 3871 copies/cell of the most abundant directly presented peptide, NP₃₆₆₋₃₇₄). Moreover, the range of IAV epitope presentation spanned only two orders of magnitude (compared to three orders by direct presentation) (Supplementary Fig. 5c). Despite this, of the 21 peptides identified by direct presentation, 15 were also detected by cross-presentation. The epitope abundance hierarchy was normalized to the amount of PB1₇₀₃₋₇₁₁ detected within each experiment (Fig. 3b). A significant correlation existed between the relative abundance of cross-presented versus directly presented peptides (RCC =

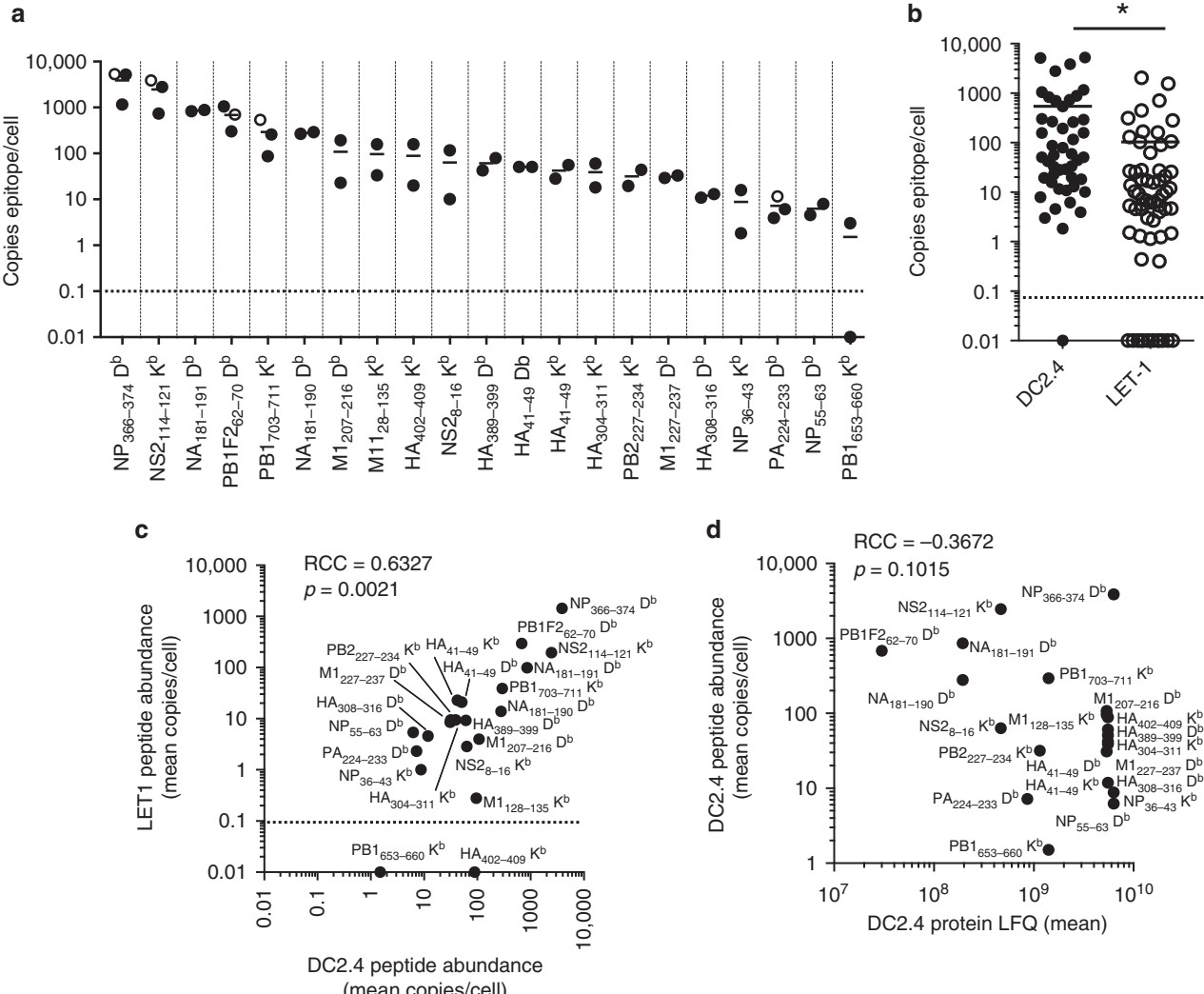

**Fig. 1** Detection and quantitation of MHCI-bound IAV peptides following direct infection. DC2.4 cells ($1 \times 10^8$) were mock treated or infected for 8 h with the PR8 strain of IAV at an MOI of 5, epitopes were eluted from immunoaffinity-purified K$^b$ and D$^b$ MHCI molecules and analyzed by LC-MRM. **a** Absolute quantitation of peptide abundance shown as peptide copies/cell. $N = 2$–3 independent infections are shown. Open circles represent initial quantitation of known epitopes. Closed circles represent quantitation of all epitopes discovered by LC-MS/MS. **b** Quantitation of peptides eluted from DC2.4 cells and LET1 cells, with each data point representing a single replicate of a particular peptide. $N = 2$–3 independent infections (*$p = 0.008$ using Mann−Whitney nonparametric test). Correlation between **c** mean peptide abundance from DC2.4 cells vs. LET1 cells ($N = 3$), and **d** mean peptide vs. relative protein abundance in DC2.4 cells, showing the Spearman's rank correlation coefficient (RCC) and associated $p$ values. Dashed lines represent the limit of detection

0.6365, $p = 0.0019$) (Supplementary Fig. 5d). However, there were some key outliers to this association. Most notably, the rank order of PA$_{224–233}$ was substantially increased by cross-presentation such that it was one of the most abundantly cross-presented peptides (Fig. 3c). Given the overall lower efficiency of the cross-presentation system, these data suggest that PA$_{224–233}$ is substantially more efficiently presented via cross- than direct presentation.

**Measurement of peptide-MHCI binding strength**. The strength of the noncovalent interaction between peptide and the binding groove of MHC has long been implicated in contributing to immunogenicity; however, the extent to which the affinity of this interaction ultimately impacts upon pMHCI abundance remains largely unknown[18]. To address the role of peptide binding affinity for MHCI in determining epitope abundance and CTL responses, for each of the peptides identified in this study we have determined peptide-MHC affinity (IC$_{50}$ nM) values experimentally

using classical competition assays[19] (Table 2). Overall, 17/20 (85%) of the peptides bound either H-2K$^b$ or D$^b$, or both with an affinity of 500 nM or better, an affinity threshold identified for the vast majority of known class I epitopes[20,21]; ten peptides can be classified as strong binders (IC$_{50} < 50$ nM) for one or the other allele, seven as intermediate binders (IC$_{50}$ between 50 and 500 nM), and three as weak (IC$_{50}$ between 500 and 5000 nM) or nonbinders (IC$_{50} > 5000$ nM). Surprisingly, there was no significant correlation between the strength of peptide binding, as defined by IC$_{50}$, and epitope abundance measured following direct or cross-presentation (Fig. 4).

**Characterization of T-cell responses to the identified IAV peptides**. The above data represent the first in-depth analysis of the differential abundance of virus peptides presented by MHC class I complexes via the routes of direct versus cross-presentation. In order to relate this information to the immunogenicity of each peptide in vivo, we systematically characterized

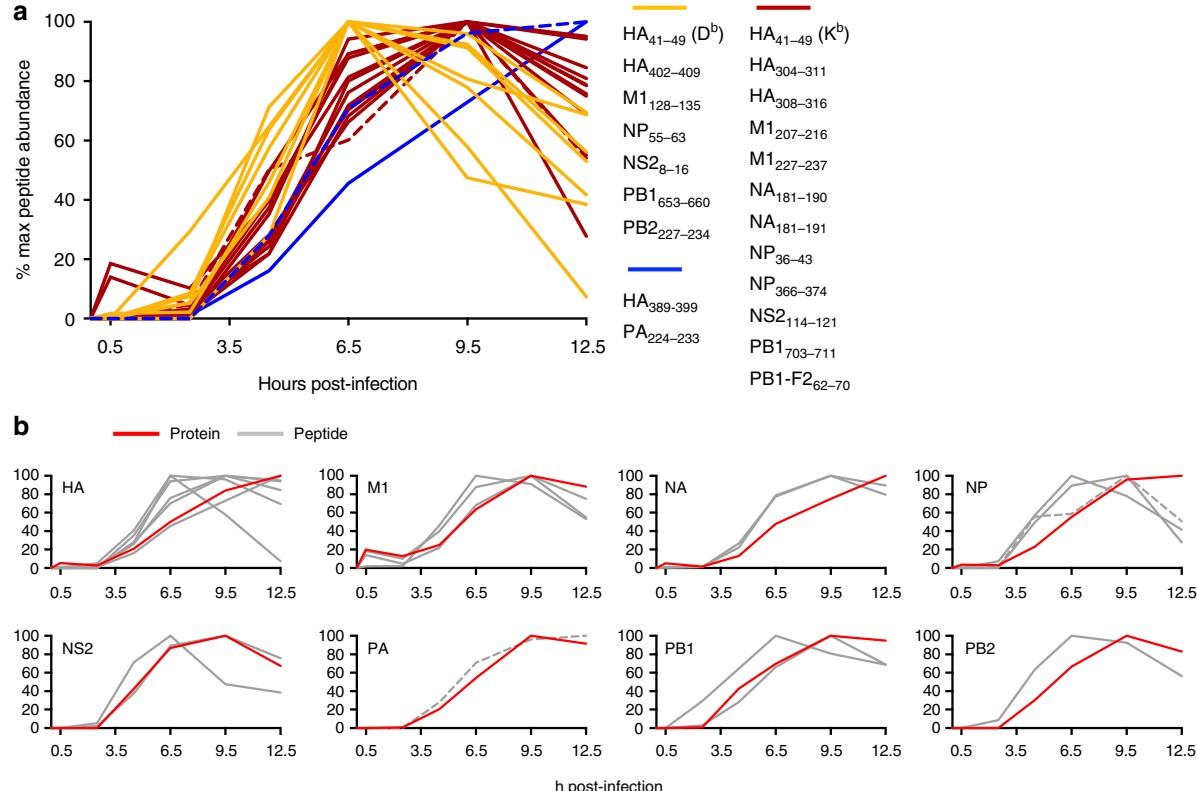

**Fig. 2** Kinetics of protein expression and peptide presentation. **a** The kinetics of peptide presentation were determined by LC-MRM analysis of MHCI-bound peptides isolated from infected DC2.4 cells at 0, 0.5, 2.5, 6.5, 9.5 or 12.5 h post infection. Data are expressed as a percentage of maximum levels detected over the timecourse. Peptides whose abundance peaked at 6.5 h are shown in yellow, those that peaked at 9.5 h shown in red, and those that continued to increase at 12.5 h are shown in blue. **b** Relative peptide abundance (gray) is plotted alongside relative protein abundance (red) kinetics for the various IAV-derived proteins. Dashed lines indicate the immunodominant $NP_{366-374}$ and $PA_{224-233}$ peptides

the immune response elicited to each peptide identified in this study after IAV infection. The magnitudes of epitope-specific CTL responses, as defined by IFN-γ production, were determined from spleen and bronchoalveolar lavage (BAL) 10 days after intranasal IAV infection of B6 mice (Fig. 5a and Supplementary Fig. 6). As shown previously, $D^bNP_{366-374}$ and $D^bPA_{224-233}$ elicited dominant CD8+ T-cell responses (comprising ~60% of the total antiviral response) (Fig. 5b, c) with all other epitopes eliciting smaller responses. Strikingly, all 20 of the peptides identified in the analysis (this assay cannot distinguish between $K^b$- and $D^b$-restricted $HA_{41-49}$-specific responses) were able to elicit a response in at least one mouse, with a consistent CD8+ T-cell response (found in ≥3 mice) detected toward 16 of the 20 IAV peptides. These public responses included all seven well-documented IAV-derived peptides ($D^bNP_{366-374}$, $D^bPA_{224-233}$, $K^bPB1_{703-711}$, $D^bPB1-F2_{62-70}$, $K^bNS2_{114-121}$, $K^bM1_{128-135}$, and $D^bNA_{181-190}$) (Fig. 5b), and a further nine peptides detected in this study. These data thus confirm that the peptides identified by mass spectrometry, despite being isolated from cells infected in vitro, are presented on MHCI during in vivo infection.

Collectively, CD8+ T-cell responses to the seven novel epitopes identified in this study comprise ~10% of the total antiviral response, as mapped thus far. Analysis of the frequency with which each peptide was able to elicit a response revealed a tendency for responses to be elicited in either a few individuals (≤ 3/11 mice; 35%) or in all mice (30%) (Fig. 5d). Perhaps unsurprisingly, the frequency with which a peptide was able to induce a CTL response was positively correlated to the magnitude of that response (Fig. 5e). These data suggest that a threshold of

immunogenicity exists, above which all mice are able to respond and below which the response is more stochastic.

**Modeling of parameters that may impact T-cell response magnitude.** Given that we have measured multiple variables that may contribute to CTL immunodominance, both in this study and earlier investigations[2,22], and obtained an epitope-based hierarchy for each of these measures, we employed an integrated statistical and modeling approach to determine their relative contributions to the CTL immunodominance hierarchy. Initially, we investigated correlations between the outcome of CTL response for each epitope and all other epitope-specific variables. For each variable, we took the mean of repeated measurements for a given epitope and computed rank correlation between variables (Fig. 6a). As observed previously, for the five epitopes for which naive CTL precursors (CTLp) have been determined, there was a negative correlation between CTLp frequency and CTL responses after infection. Despite strong correlations in virus protein levels between DC2.4 and LET1 cells, protein abundance showed weak association with epitope abundance via direct or cross-presentation, nor did it correlate with CTL response magnitude. The variables found to be significantly positively associated with CTL response magnitude were the peptide abundances driven by each of the presentation pathways analyzed. Surprisingly, while $IC_{50}$ showed a moderate correlation (RCC = −0.4) with the CTL response, it showed poor associations with all measures of epitope abundance.

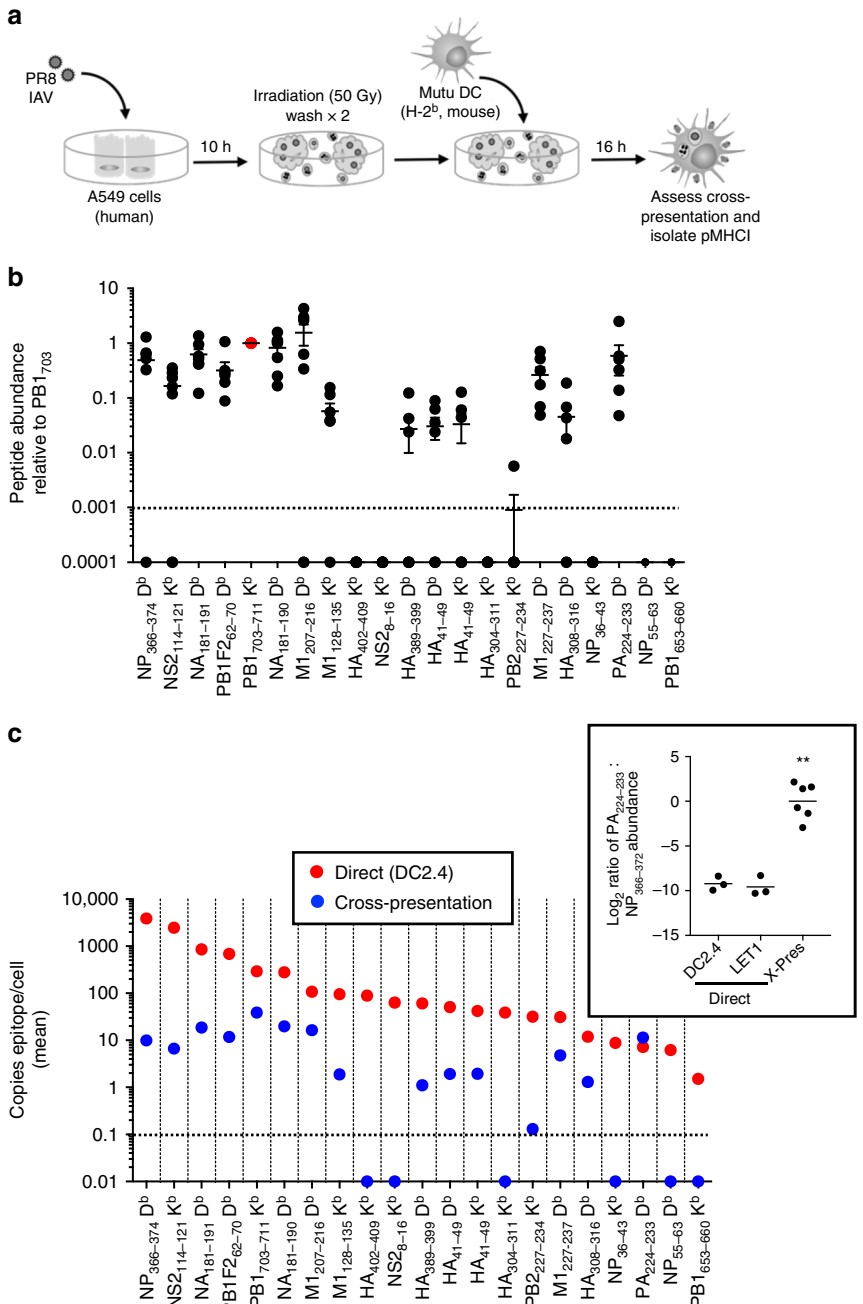

**Fig. 3** Detection and quantitation of cross-presented IAV peptides. **a** Schematic of in vitro cross-presentation workflow. A549 cells ($5 \times 10^7$) were infected for 10 h with PR8 IAV at an MOI of 10. The A549 cells were then γ-irradiated (50 Gy) and co-incubated for 16 h with $5 \times 10^7$ Mutu DCs. Epitopes were then eluted from immunoaffinity-purified $K^b$ and $D^b$ MHCI molecules and analyzed by LC-MRM. **b** Shown is the mean relative abundance ± SEM of each peptide compared to $PB1_{703-711}$ (in red) within the same experiment. **c** Absolute quantitation (mean epitope copies/cell shown) of peptide abundance following direct infection and cross-presentation. Direct epitope abundance shown in red circles while cross-presented epitope abundance shown in blue circles. Inset shows the ratio of $\log_2$ transformed $PA_{224-233}$: $NP_{366-374}$ epitope abundance for each of the different analyses of epitope presentation. $N = 7$ independent cross-presentation assays. **p < 0.0001 comparing cross-presentation and direct (DC2.4) ratios, and cross-presentation and direct (LET1) ratios, using one-way ANOVA with Tukey post-hoc test

To investigate the relationship between the antiviral T-cell response and the different epitope-related quantities in more detail, we moved from an analysis of rank correlations to an analysis of actual numeric values for each variable. We performed simple linear regression to evaluate linear correlations on the log scale between CTL response and each individual predictor variable. This analysis showed linear correlations of epitope presentation (DC2.4 and LET1 direct presentation and cross-presentation) and T-cell response magnitude, and no correlation

between T-cell response and protein levels. This analysis also found that peptide $IC_{50}$ was significantly correlated with T-cell response magnitudes (Fig. 6b).

These findings suggested that a multivariate linear model might further improve the ability to predict CTL magnitude. To test this, we used an exhaustive subset selection approach with cross-validation to assess multivariate linear regression models with all possible combinations of variables to determine their ability to predict CTL magnitude, as quantified by $R^2$. Naïve CTLp

frequency was excluded from this analysis due to the limited dataset. The best-performing model only included two variables: cross-presentation and $IC_{50}$, and had a cross-validated $R^2$ of 0.68 (and an $R^2$ of 0.76 without cross-validation), indicating a substantial improvement over any individual predictor (Fig. 6b). Using a proportional marginal variance decomposition (PMVD) method, we found that around 66% of the relative importance was attributed to cross-presentation, with the remaining 34% attributed to $IC_{50}$. This finding is in line with the univariate analysis shown in Fig. 6b, which identified cross-presentation as the strongest contributor to CTL response magnitude. While the direct presentation variables also showed strong correlation in the univariate analysis, they are strongly correlated with cross-presentation and are thus not selected by the model building routine.

Next, to investigate the role of each individual epitope, we performed the above analysis while withholding a different epitope from the data each time. For 19 of the 20 epitopes

removed, all best-fit models included the variables cross-presentation and $IC_{50}$ as the factors that contributed the most to CTL response magnitudes. Intriguingly, removal of the $PA_{224-233}$ epitope changed the best-fit model to one that included the DC2.4 direct presentation and the DC2.4 and LET1 protein level variables. Thus, it seems that cross-presentation is the major correlate for the $D^bPA_{224-233}$-specific CTL response, and that its weighting as a superior global correlate (compared to direct presentation) for the CTL response was due to its influence on the $D^bPA_{224-233}$-specific CTL response.

Finally, to determine if a model that allowed more complex relations (i.e. beyond linear) between CTL magnitude and epitope-specific variables would perform better, we applied a support vector machine model (SVM), a random forest (RF), and a gradient boosted regression tree model (GBM). We also used a LASSO method as an alternative way of doing variable selection for a linear model. However, none of these models provided better performance than the multivariate linear model (measured by cross-validated $R^2$) after tuning and training. Further details are provided in the Supplementary Methods. Thus, for the dataset analyzed here, cross-presentation and the $IC_{50}$ variables additively determine T-cell response strength, with no further predictive strength gained from the other variables or from a more complicated model structure.

## Discussion

Mass spectrometry analyses of the nature and abundance of IAV-derived peptides presented on MHCI after infection identified 21 peptides (including seven novel epitopes), all of which were able to elicit $CD8^+$ T-cell responses after infection. Of the 21 peptides identified in this study, seven were 10–11 aa in length, including four of the seven novel peptides. Notably, all of these longer IAV-derived peptides were bound by H-2D$^b$ while all seven of the shorter octamers were bound by H-2K$^b$, consistent with previous observations that MHC alleles exhibit distinct peptide length preferences[23,24]. The identification of 10–11 aa peptides is also consistent with the observation that a substantial proportion of naturally occurring mouse MHCI-bound peptides are longer than the canonical length of 8–9 aa. Such observations are becoming more apparent as global immunopeptidomics analyses increase[18,25], and underscore the potential deficiencies in many predictive algorithms[26]. That long peptides from viruses and tumors are naturally processed and presented on MHCI suggests they likely play a key role in antiviral and antitumor immunity[25]. Long peptides have been found to be accommodated by MHCI

| Table 2 Measured $IC_{50}$ (nM) of each peptide | | |
|---|---|---|
| | **H-2D$^b$** | **H-2K$^b$** |
| $NP_{36-43}$ | 3303 | **1.9** |
| $NP_{55-63}$ | **184** | 463 |
| $NP_{366-374}$ | **18** | — |
| $PA_{224-233}$ | **0.16** | 38 |
| $PB1_{653-660}$ | — | **156** |
| $PB1_{703-711}$ | 28988 | **1.9** |
| $PB1\text{-}F2_{62-70}$ | **18** | 12403 |
| $PB2_{227-234}$ | 20903 | **2.8** |
| $NS2_{8-16}$ | 10687 | **185** |
| $NS2_{114-121}$ | — | **4.7** |
| $M1_{128-135}$ | — | **92.6** |
| $M1_{207-216}$ | **11225** | 26943 |
| $M1_{227-237}$ | **5285** | 3306 |
| $HA_{41-49}$ | 872 | **7.9** |
| $HA_{304-311}$ | — | **4.7** |
| $HA_{308-316}$ | **1971** | 1138 |
| $HA_{389-399}$ | **118** | 22004 |
| $HA_{402-409}$ | 6180 | **3.0** |
| $HA_{467-476}$ | **173** | — |
| $NA_{181-190}$ | **65** | 262 |
| $NA_{181-191}$ | **64** | 585 |

Data for each peptide's restricting allele are shown in bold type. A dash indicates $IC_{50} > 40,000$ nM

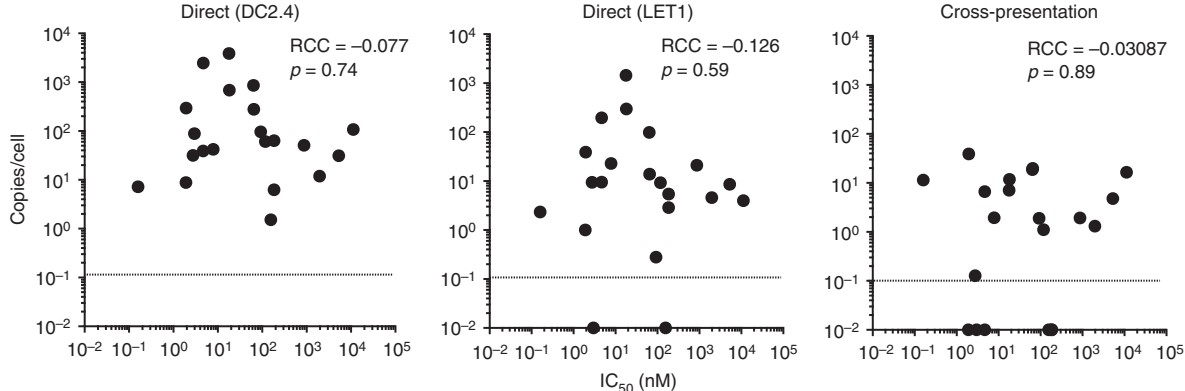

**Fig. 4** Correlation of peptide-MHCI binding affinity and epitope abundance. The mean peptide abundance as determined following direct infection of DC2.4 cells or LET1 cells or via A549 × Mutu cross-presentation was correlated with the mean $IC_{50}$ of each bound peptide, showing the Spearman's rank correlation coefficient (RCC) and associated *p* values

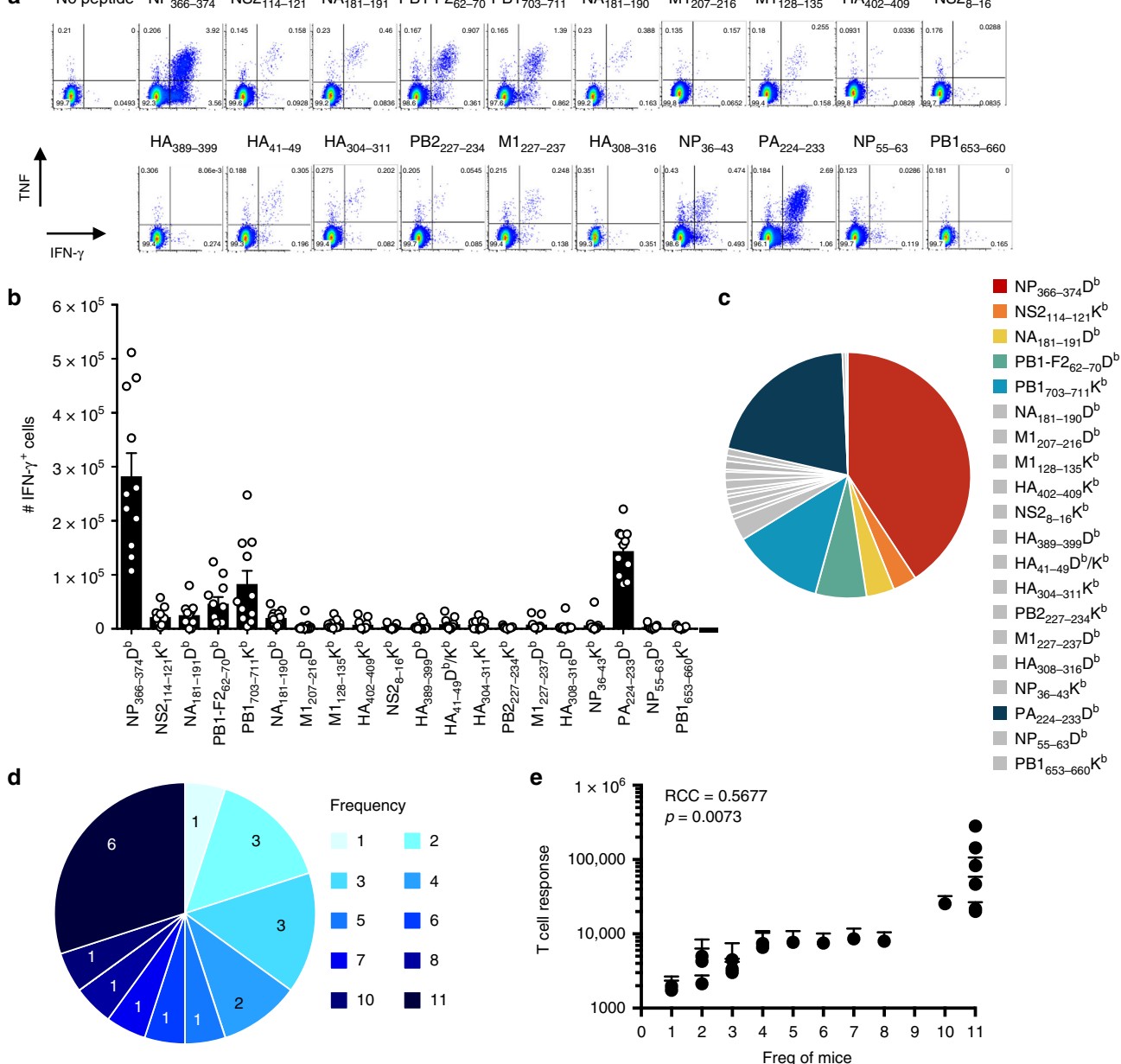

**Fig. 5** Analysis of primary CD8+ T-cell responses to IAV-derived peptides. Naive B6 mice were infected i.n. with 1000 pfu PR8 IAV, and spleens harvested 10 days later. Splenocytes, enriched for CD8+ T cells, were stimulated for 5 h in the presence or absence of 1 μM of individual peptides, and analyzed by intracellular staining for the production of IFN-γ and TNF. **a** Representative dot plots of CD8+ T-cell production of IFN-γ and TNF following peptide re-stimulation. Values indicate the percentage of CD8+ cells within each gate. **b** Number of splenic CD8+ IFN-γ+ cells responding to each peptide from 11 mice in two independent experiments ± SEM. **c** Relative contribution of each detectable epitope-specific CD8+ T-cell response to the total detectable response. **d** Frequency (out of a total of 11 mice) with which the identified peptides are able to elicit CD8+ T-cell responses. Numbers within pie chart indicate number of peptides. **e** The mean + SEM CD8+ T-cell response plotted against the cumulative number of mice in which the responses were detected. $N = 11$ mice from two independent experiments. Source data are provided as a Source Data File

via a central bulging from the antigen binding cleft or occasionally via amino- or carboxy-terminal extensions from the MHCI groove[25,27]. Structural analysis of TCRs binding long peptides in complex with MHCI show that TCRs can achieve pMHCI recognition using a variety of strategies, including flattening the bulged peptide to increase MHCI contact, or sitting atop the peptide making minimal MHCI contact[28]. Collectively, given their prevalence and the demonstrated ability of T cells to recognize long peptides, our data further suggest that proteome-wide discovery-based approaches should now be utilized to obtain a comprehensive representation of the peptidome.

Whilst we have conceivably captured all available H-2D$^b$ and K$^b$ complexes from cells through immunoprecipitation, there is potentially some bias in peptide identification. For example, C$_{18}$ chromatography may miss overly hydrophilic or hydrophobic sequences and, due to the stochastic nature of conventional MS modalities for peptide isolation and fragmentation, coupled to protein-centric search algorithms, we may fail to detect and assign peptides of lower abundance, poor ionization, or where ambiguous spectra have been acquired[29]. It is also conceivable that a fraction of the viral repertoire is presented through proteasome-catalyzed peptide splicing, an area that requires

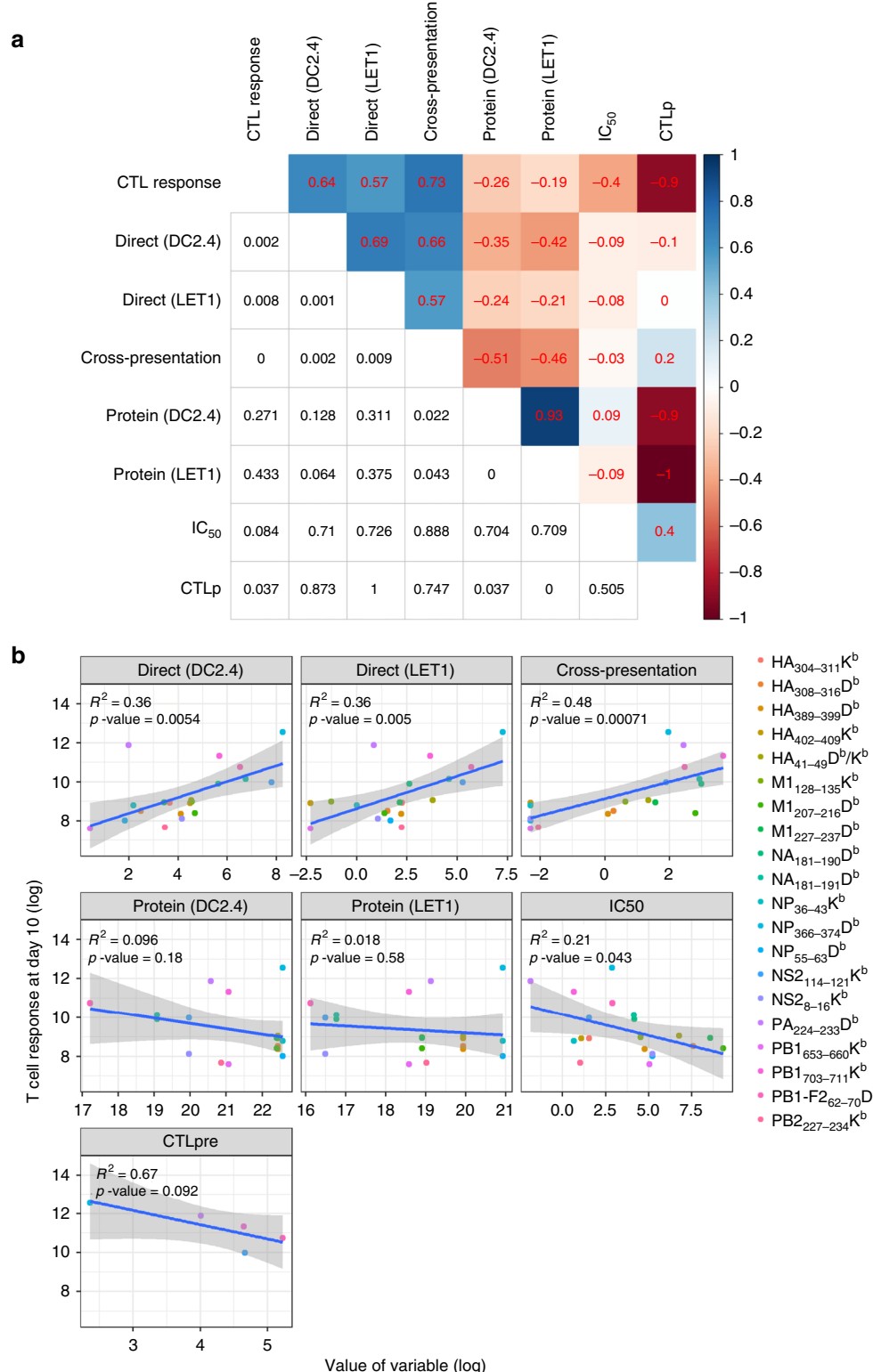

**Fig. 6** Analysis of parameter contribution to CD8$^+$ T-cell response hierarchy. **a** Correlations as assessed through spearman rank correlation test between all variables. Spearman rank correlation coefficient is shown in the upper diagonal and associated $p$ value (not adjusted for multiple comparison) in the lower diagonal. Color coding indicates strength of positive (blue) or negative (red) correlation between specific variables. Note that because the contribution of HA$_{41-49}$ bound to H-2D$^b$ or H-2K$^b$ could not be distinguished for CTL response, the abundance of HA$_{41-49}$ bound to H-2D$^b$ and H-2K$^b$ have been added for each of the presentation modalities to yield a single value. **b** Linear correlations between log-transformed outcome (CTL response magnitude at day 10) and log-transformed predictor variables. For each variable, we took the mean of all measurements before log transformation. $R^2$ values are shown for the performance of the model on the actual data used to fit the model. CTLp frequencies for five epitopes taken from ref. [22]

future investigation now that accessible algorithms are available[30].

In this study we comprehensively characterize epitope presentation during direct and cross-presentation and demonstrate its influence on CTL immunodominance hierarchies after IAV infection. This builds on a previous study that has shown that CTL precursor frequency fails to predict the magnitude of the ensuing immune response[22], making this an ideal model in which to define the impact of epitope abundance on antigenicity and CTL response magnitude. Following, we highlight several important findings that arise from this direct analysis of antigenicity and the implications for CTL recruitment and expansion during infection. Firstly, we observed that directly infected cells show the same rank order of epitope presentation and abundance despite their tissue of origin. Given that DC2.4 cells express immunoproteasome subunits[31], this challenges the notion that the immunoproteasome generates a unique repertoire of MHCI-binding peptides to the constitutive proteasome by virtue of unique cleavage site specificity[32–34]. Our findings support later studies, which showed a positive effect of immunoproteasome on the abundance of antigenic peptides, but little evidence for the generation of a qualitatively distinct peptide repertoire[35,36]. Secondly, our mass-spectrometry-based approach identified a number of new IAV peptides that were all immunogenic in at least one mouse, and that for many subdominant epitopes there was a private response where recruitment of these CTL specificities into the immune repertoire appears to be more stochastic than the responses to the more dominant T-cell determinants. The most plausible explanation for poor or sporadic responses despite detectable epitope presentation is a lack of available naïve T cells. So-called holes in the T-cell repertoire have been attributed, in part, to the deletion of cross-reactive T cells due to excessive similarity to self-peptides[37]. Alternatively, some epitopes elicit poor T-cell responses despite a large number of available naïve T cells. In this case, the subdominance of the response may be a consequence of poor naïve T-cell quality, due either to a low affinity TCRs[38] or noncanonical TCR−pMHC interactions that are incapable of driving robust signaling[39]. Such observations highlight the need to rethink how we define immunogenicity and reflect the complex nature of the immune response that are not uniform even in syngeneic experimental systems. Thirdly, we observed that while direct presentation yielded peptide abundances that spanned three orders of magnitude, the levels of cross-presented IAV-derived epitopes were much more uniform. This suggests that cross-presentation normalizes antigen presentation, providing a niche for T-cell priming that facilitates the expansion of T-cell clonotypes on a level playing field with respect to epitope abundances. Finally, and remarkably, when comparing direct versus cross-presentation we observed that although optimal presentation of $NP_{366–374}$ was driven by direct presentation, the unique capacity of cross-presentation to drive high-level display of $PA_{224–233}$ was associated with its immuno-dominant status. This was particularly striking in light of the exceedingly poor presentation of $PA_{224–233}$ via direct presentation (ranking 19th of 21 peptides). The disparate behavior of these two immunodominant epitopes highlights the difficulty in predicting CTL response magnitude and confounds attempts to model the evolution of viral immunity.

Previous studies have indicated that the characteristics of antigens that are efficiently cross- versus directly presented are contrasting—short-lived, unstable cytosolic proteins are optimally presented via the direct route, while more stable forms of antigen, such as cell-associated proteins, are preferred for cross-presentation[40–42]. Evidence for this comes from minigene expression of three IAV-derived H-2D$^b$-binding peptides, $NP_{366–374}$, $PA_{224–233}$, and $PB1-F2_{62–70}$, which demonstrated the

unique capacity of the $PA_{224–233}$ peptide to be retained as a long-lived cytosolic pool of peptide that was able to sustain presentation for hours after termination of protein synthesis[43]. Collectively, our findings go some way in demystifying the relative contributions of direct and cross-presentation by demonstrating that peptides largely have access to both pathways. However, our data also support the notion that distinct peptide characteristics can markedly favor cross-presentation over direct presentation, and such distinctions can have major effects on the CTL response hierarchy.

In the current analysis of IAV infection, in contrast to vaccinia virus (VACV) infection of the same DC2.4 cell line[4], we noted only minor variation in the kinetics of protein expression and no obvious correlation between that and the presentation kinetics of directly presented peptides sourced from those proteins. Of course, VACV, unlike IAV, is a large virus that exhibits temporal regulation of protein expression, which is likely to more easily facilitate the observation of associations between protein expression and epitope presentation that were not detected here. We did observe rapid (within 30 min) presentation of epitopes derived from relatively abundant viral structural proteins (M, NA, HA), that were likely derived from pre-existing viral proteins rather than de novo production[44]. For later timepoints, (≥3 hpi) the peak presentation of all peptides either preceded or was coincident with peak protein expression, suggesting that antigen presentation is linked with the translation of newly synthesized polypeptides. This would be consistent with any mechanism that allows for the sampling of newly translated proteins, including the disposal of defective products of translation or those that fail to achieve their final function in the viral lifecycle[45]. This is also supported by other studies demonstrating the rapid presentation of peptides after translation, and overall poor correlation between pMHCI levels and protein turnover[4,46].

Peptide affinity for MHC has been shown in several systems, including VACV and lymphocytic choriomeningitis virus (LCMV), to correlate with the immunogenicity of pMHC during infections[47,48]. The elevated affinity is typically thought to increase the abundance of pMHC complex on the cell surface. Surprisingly, and in accordance with our recent findings for VACV[18], our data revealed that the strong correlation between peptide affinity for MHCI (as measured by $IC_{50}$) and T-cell response magnitude occurred independently of epitope abundance. This is exemplified by the similar relative levels of the $HA_{41–49}$ peptide presented on H2D$^b$ and K$^b$, following both direct and cross-presentation, despite this peptide exhibiting a substantially greater affinity for K$^b$. Additionally, the affinities of the $NP_{366–374}$ and $PA_{224–233}$ peptides for H2D$^b$ were the highest observed, corresponding with their immunodominant response status, but not with their differential abundance following direct presentation. Thus, in the absence of a notable effect on epitope abundance, it is possible that a relatively low affinity peptide/ MHC interaction, under the tensile force that is characteristic of agonist TCR-pMHCI recognition[49], leads to unstable interactions between the TCR and pMHC that result in suboptimal T-cell activation. It is also possible that peptide affinity for MHC primarily influences the longevity of pMHC complexes, which is also proposed to be a key indicator of antigenicity[50,51], independently of the relative abundance of those complexes at a given timepoint.

The relevance of this study to the immune CD8$^+$ T-cell responses in humans should be highlighted, especially since human responses to many pathogens, and IAV in particular, reflect recall, rather than primary, responses. Nonetheless, humans must acquire T-cell immunity to viruses, and the primary CTL response to the initial exposure event establishes a framework from which an individual's capacity to respond to

virus rechallenge thereafter is defined. For example, studies suggest that repeated exposures to the same virus tend to focus responses that were established following primary infection[52,53], and that in some cases pre-existing memory populations may detrimentally impact subsequent CD8+ T-cell responses[54,55]. Thus, an understanding of the key contributors to antigenicity and immunodominance in primary CD8+ T-cell responses is imperative for a comprehensive understanding of both primary and recall CD8+ T-cell responses in humans. Certainly, characteristics of mouse and human antiviral CD8+ T-cell responses (reproducible immunogenicity and immunodominance in MHC-matched individuals) are similar, and direct analyses of epitope presentation are becoming highly feasible in human cells[56]. Such studies will ultimately elucidate whether similar mechanisms drive CTL immunodominance in the human setting.

In summary, our study provides new insights into the mechanisms by which antigen presentation influences CTL response magnitude after infection, with both direct and cross-presentation making pivotal contributions to distinct epitope-specific CTL responses. Such information, in conjunction with further studies relating the direct quantitation of epitope abundance to immune outcomes, are essential for the understanding and informed development of vaccines and immunotherapies targeting T-cell responses.

## Methods

**Cell lines**. The two murine cell lines used in direct infection were DC2.4[10], a DC-like cell line derived from B6 mice (kindly provided by Dr. Ken Rock, University of Massachusetts, Worcester, MA), and LET1 cells, a type I lung epithelial cell line derived from B6 mice[11]. The cell lines used in cross-presentation were the human adenocarcinomic alveolar basal A549 cells (ATCC #CCL-185) and the murine CD8α+ Mutu DC cells[16] (kindly provided by Dr. Hans Acha-Orbea, University of Lausanne, Epalinges, Switzerland). All the cells were cultured at 37 °C and in 5% $CO_2$ in recommended growth medium (purchased from Life Technologies) supplemented with 10% fetal bovine serum (FBS), 50 IU/ml penicillin, 50 μg/ml streptomycin and 2 mM L-glutamine. DC2.4, LET1 and A549 cells were cultured in DMEM. Mutu cells were grown in IMDM supplemented with Gluta-MAX™. Murine EL-4 thymoma cell lines were maintained in RPMI-1640 supplemented with 10% fetal FBS, 50 IU/ml penicillin, 50 μg/ml streptomycin and 2 mM L-glutamine (RF-10) and EG7, a subclone of EL-4 transfected with ovalbumin (OVA) gene[57], was maintained in RF-10 supplemented with 500 μg/ml G418 (Invitrogen).

**Mice and viruses**. C57Bl/6 (B6, H-2^b) WT mice and OT-1 transgenic mice expressing a TCR specific for the OVA$_{257-264}$ SIINFEKL peptide in complex with H2-K^b were housed in specific pathogen-free animal facility of the Department of Microbiology and Immunology, University of Melbourne (Victoria, Australia) or at the Animal Research Laboratories (ARL) at Monash University. Mice aged at 6–12 weeks and PR8 H1N1 IAV (A/Puerto Rico/8/34) and PR8-OVA IAV[17] were used in this study. All animal experimentation was conducted following the Australian National Health and Medical Research Council Code of Practice for the Care and Use of Animals for Scientific Purposes guidelines for housing and care of laboratory animals and performed in accordance with Institutional regulations after pertinent review and approval by the University of Melbourne and Monash University Animal Ethics Committees.

**Synthetic peptides**. Isotope-labeled peptides (>90% purity) were purchased from Mimotopes Pty Ltd (Clayton, Victoria, Australia) and dissolved in 100% (v/v) dimethyl sulfoxide or 20~30% (v/v) ACN according to the manufacturer's instructions. The concentration of each peptide was determined using a Direct Detect® Infrared Spectrometer (Merck, Germany).

**Direct infection of cells**. DC2.4 or LET1 cells were allowed to grow to 80–90% confluence in T175 flasks and harvested by trypsinisation. Cells (~1 × 10^8) were washed and mock- or IAV-infected at a MOI of 5 at 37 °C for 1 h. The cells were then incubated for a further 7 h at 37 °C with gentle rolling, and counted. Cells were washed, snap frozen and stored at −80 °C.

**Infection efficiency assay**. At 8 hpi IAV or mock-infected DC2.4 or LET1 were fixed, permeabilized and stained with fluorescein isothiocyanate (FITC)-conjugated anti-influenza A nucleoprotein (NP) monoclonal antibody (D67J, eBioscience, San Diego, CA) for 30 min at 4 °C, followed by flow cytometric acquisition on a

FACSCanto (BD Biosciences, Oxford, UK) and analysis using FlowJo software (Tree Star Inc, Ashland, OR).

**Intracellular cytokine staining**. Lymphocytes were obtained from the pneumonic lung by bronchoalveolar lavage (BAL) and adherent cells were removed by incubating on plastic for 1 h at 37 °C. Single-cell preparations of spleen were enriched for CD8 T cells by panning on tissue culture plates coated previously with AffiniPure goat anti-mouse IgG + IgM (H + L) (Jackson ImmunoResearch Labs). CD8+ T-cell responses to each candidate epitope were tested via intracellular cytokine staining. Enriched lymphocytes from spleens or bronchoalveolar lavage of mice infected 10 days previously with IAV were incubated with 10 U/ml IL-2 (Roche) and 1 μg/ml Golgi-plug (BD Biosciences) in the presence or absence of 1 μM synthetic peptide in 96-well plates and cultured for 5 h at 37 °C and 5% $CO_2$. Cells were then stained for surface expression of CD8α and intracellular IFN-γ and TNF. Data was acquired by flow cytometry (FACSCanto, BD Biosciences) and analysis was performed using Flowjo version 9.6 (FlowJo LLC, Ashland). Single viable live CD8+ lymphocytes were gated for analysis (Supplementary Fig. 7). A response was deemed positive if the test response (with background subtracted) was equal to or greater than 2× the average background values.

**In vitro cross-presentation**. A549 cells (antigen donor cells) were mock-infected or infected with IAV virus or IAV-OVA virus (MOI = 10) for 10 h prior to γ-irradiation (50 Gy) to inhibit the proliferation of viruses and induce apoptosis of the cells. After washing, the irradiated cells were incubated with Mutu cells (GFP+) at a ratio of 1:1 or 1:2 in the presence of 500 nM CpG oligodeoxynucleotides (InvivoGen) at 37 °C and harvested at the designated time. For preliminary analysis of cross-presentation, the Mutu cells were then co-cultured with VPD450-labeled OT-1 CD8+ T cells at ratio of 1:1 at 37 °C for 24 h. OT-1 T-cell proliferation was assessed based on the VPD450 dye dilution in CD8+ T cells. Alternatively, the Mutu cells were stained with monoclonal 25-D1.16 antibody, specific for the H-2K^b bound peptide OVA$_{257-264}$ (SIINFEKL), followed by a PE-conjugated anti-mouse IgG antibody (BD Biosciences), and analyzed by flow cytometry.

**Purification of MHCI-peptide complexes**. Frozen cell pellets of murine cells (DC2.4s, LET1 or A549/Mutu cells) were lysed in 5 ml of lysis buffer containing 50 mM Tris pH 8.0, 150 mM NaCl, 0.5% IGEPAL and protease inhibitors. The lysates were incubated at 4 °C for 1 h under rotation and centrifuged at 16,000 × g for 10 min. After centrifugation, the MHC/peptide in the supernatant were captured sequentially by immunoaffinity purification using columns containing protein A sepharose (GE Healthcare) conjugated to firstly Y-3 (anti-H-2K^b) and then 28-14-8s (anti-H-2D^b) monoclonal antibody[58]. The bound complexes were then eluted with 10% acetic acid and mixed with 50–200 fmol of each isotopic peptide. The mixture of peptides and dissociated MHCI molecules were further fractioned on a C18 reverse-phase column (5 μm, 50 × 4.6 mm I.D., Chromolith Speed Rod, Merck) on an ÄKTAmicro HPLC system (GE Healthcare) across an increasing gradient of buffer B (80% acetonitrile, 0.1% TFA in water) at a constant flow rate of 1 ml/min. Fractions were vacuum concentrated (Labconco Centrivap) and resuspended in 0.1% formic in water to a volume of 20 μl prior to mass spectrometry analysis.

**Identification of viral peptides by IDA mass spectrometry**. Peptide fractions were analyzed by LC-MS/MS with trapping and separation using an Eksigent nanoLC-Ultra 2D plus system (SCIEX) combined with a cHiPLC-nanoflex system (SCIEX) in trap-elute mode (trap column: 200 μm × 0.5 mm ChromXP C18-CL 3 μm, 120 Å; analytical column: 75 μm × 15 cm ChromXP C18-CL 3 μm, 120 Å). Separation was achieved at a flow rate of 300 nl/min with increasing linear concentrations of buffer B (80% acetonitrile, 0.1% formic acid in water). Acquisition of spectra was by a TripleTOF® 5600+ (SCIEX) mass spectrometer through an uncoated silica PicoTip™ nano electrospray emitter (New Objective Woburn, MA) with an ion spray voltage of 2300 V in positive ion mode. The MS analysis was performed in Information Dependent mode (IDA) using the following parameters: 200 ms of MS1 scan acquired from 200 to 1800 Da m/z followed by 150 ms MS/MS scan over an m/z range of 60–1800 Da. Up to 20 of the most intense ions with a charge-state of +2 to +5 were selected for MS/MS per cycle if they exceeded 40 counts per second (cps), and then they were excluded for further analysis for 30 s after two occurrences. Data analysis was performed on ProteinPilot version 4.5 (SCIEX). An in-house database was used which contained protein sequences of *Mus musculus*, influenza H1N1 IAV-PR8 strain and proteins from the alternative open reading frames of IAV-PR8. The peptides generated from the software were exported into Microsoft Excel for further data analysis at a 5% false discovery rate cut-off.

**Identification of viral peptides by LC-MRM**. LC-MRM was initially employed to detect peptides reported in previous studies. Initially, a list of viral peptides was generated by searching the open-source immune epitope database and analysis resource[59]. The design of MRM transitions for each peptide were aided by the Skyline software[60]. The positive double charged precursor ion and at least three single or double charged product ions (y and b ions) for each peptide were selected.

The collision energy (CE) and declustering potential (DP) was calculated automatically in Skyline according to the rolling collision energy equations used by the QTRAP® 5500 (SCIEX). The dwell time was set to 5 ms to ensure sufficient data points (at least eight) could be acquired across the chromatographic peak. The identification of peptides using LC-MRM MS was performed as follows: separation of the peptides was performed on the same Eksigent nanoLC system as described above; after online LC separation, the MS was operated in a nonscheduled MRM mode followed by an Enhanced Product Ion (EPI) scan triggered each cycle for the most intense MRM transition. The MRM transitions were acquired at unit resolution in the first and third quadrupoles (Q1 and Q3). In EPI scanning mode, rolling collision energy (CE) was applied to acquire fragment ion spectra for selected precursor ions. The results were then analyzed by Skyline.

**Design of viral peptides MRMs for peptide quantification**. The design of MRM transitions for the quantification of identified influenza peptides was performed as follows: initially, the predominant precursor ion of each peptide was selected by Q1 in EMS scanning mode and the product ion spectra were obtained in EPI scanning mode. At least four of the most intense precursor-product ion pairs for each peptide were chosen to generate MRM transitions. The CE for each transition was further optimized by ramping CE across a range around the instrument-predicted value. Two MRM methods were established: nonscheduled MRM and scheduled MRM, the latter for large numbers of transitions (>500). In nonscheduled MRM mode, the dwell time for each transition was set to 5 ms. In scheduled MRM mode, the transitions of each peptide were only acquired at the expected retention time using a 300 s MRM detection window and a target cycle time of 3 s. An EPI scan was also conducted.

**Analysis of IC$_{50}$**. Classical competition assays to quantitatively measure peptide binding to the mouse class I H-2 K$^b$ and D$^b$ molecules were based on the inhibition of binding of high affinity radiolabeled peptides to purified MHC molecules[19]. Briefly, 0.1–1 nM of radiolabeled peptide was co-incubated at room temperature with purified MHC in the presence of a cocktail of protease inhibitors. Following a 2-day incubation, MHC-bound radioactivity was determined by capturing pMHCI complexes on Lumitrac 600 plates (Greiner Bio-one, Frickenhausen, Germany) coated with either the Y3 (anti-H-2 K$^b$) or 28-14-8s (anti-H-2D$^b$, L$^d$ and D$^q$) monoclonal antibodies, and bound cpm measured using the TopCount (Packard Instrument Co., Meriden, CT) microscintillation counter. The concentration of peptide yielding 50% inhibition of binding of the radiolabeled peptide was calculated and, under the conditions utilized, where [label] < [MHC] and IC50 ≥ [MHC], measured IC$_{50}$ values are reasonable approximations of true K$_d$[61]. Each competitor peptide was tested at six different concentrations covering a 100,000-fold range, and in three or more independent experiments. As a positive control, the unlabeled version of the radiolabeled probe was also tested in each experiment.

**Relative quantitation of viral protein expression**. Lysate from flow-through immunoprecipitation experiments was reduced through treatment with TCEP (tris (2-carboxyethyl)phosphine) at a final concentration of 5 mM for 30 min at 60 °C. Protein extraction was carried out by loading the sample onto a filter-assisted sample preparation (FASP) column[62], alkylating cysteine residues through treatment with 50 mM iodoacetamide for 20 min at room temperature in the dark and digestion of proteins with trypsin (enzyme:protein ratio of 1:100) at 37 °C overnight. Digested peptides were eluted from the column with 50 μl of 0.5 M sodium chloride and further purified by C$_{18}$ tips (OMIX) prior to LC-MS analysis. For all runs, an online Eksigent nanoLC-Ultra 2D Plus was used, equipped with a cHiPLC-nanoflex system. A trap column (200 μm × 0.5 mm ChromXP C18-CL 3 μm 120 Å) was used for sample loading at 5 μl/min in 100% buffer A, followed by an analytical column (75 μm × 15 cm ChromXP C18-CL 3 μm, 120 Å) operating at 300 nl/min under the following gradient conditions: 2–43% buffer B over 120 min, 43–98% B over 1 min, hold at 98% for 4 min and then a decrease back to 2% B over 1 min, holding for another 9 min at 2% B until the end of the run. For SWATH-MS analysis of the DC2.4 infection timecourse, an initial spectral library of tryptic peptides was generated by subjecting each sample to data-dependent acquisition on a 5600$^+$ TripleTOF® system (SCIEX) using the following parameters: 200 ms MS1 scan acquisition from 200 to 1800 m/z followed by 120 ms MS2 acquisition scan from 60 to 1800 m/z. Acquired spectra were searched against the combined IAV and mouse proteome database (Uniprot; March 2015) using ProtoinPilot™ (SCIEX) and the resulting combined search file used for spectral library generation in Skyline. For SWATH-MS acquisition, the following acquisition scheme was used: 250 ms MS1 scan across 300–1800 m/z, followed by 28 sequential SWATH windows of 25 Da each (1 Da overlap) from 300 to 1000 m/z. For each window, MS2 spectra were acquired for 100 ms across a scan range of 100–1800 m/z.

For LFQ analysis across direct infection of DC2.4 and LET-1 cells, or cross-presentation in Mutu cells, data were obtained from a Q Exactive™ Plus (Thermo Scientific) mass spectrometer, coupled online with an RSLC nano-LC system (Ultimate 3000, Dionex). Tryptic peptide digests corresponding to 2 μg of material were loaded onto an Acclaim PepMap 100 trap column (100 μm × 2 cm, nanoViper C$_{18}$, 100 Å pore size; Thermo Scientific) in buffer A at a flow rate of 15 μl/min. Peptides were separated across an RSLC PepMap 100 C18 nano column (75 μm × 50 cm, 3 μm, 100 Å pore size; Thermo Scientific) at a flow rate of 250 nl/min under

the following gradient conditions: 0–2 min hold at 2.5% buffer B, then 2.5–7.5% B from 2 to 3 min, then 7.5–40% B from 3 to 123 min, 40–99% B from 123 to 128 min, hold at 99% B from 128 to 134 min, then decrease of 99–2.5% B over 1 min, followed by re-equilibration at 2.5% B for 20 min until the end of the run. The Q Exactive™ Plus was operated in data-dependent acquisition mode, with an MS1 scan acquired across a mass range of 375–1800 m/z at a resolution of 70,000 at 200 m/z. Automatic Gain Control (AGC) target was set to a value of $3 \times 10^6$ and maximum ion injection time (IT) of 50 ms. Dynamic exclusion was set to 20 s. The 12 most intense multiply charged ions were sequentially isolated and fragmented in the octopole collision cell by higher-energy collisional dissociation (HCD) with the following parameters: 17,500 resolution, MS2 AGC target value of $2 \times 10^5$, normalized collision energy (NCE) of 27% and maximum IT of 120 ms and 1.8 m/z isolation window. Data searching was carried out using MaxQuant[12] (version 1.5.2.8) against a combined database of *Mus musculus*, *Homo sapiens* and influenza strain PR8 (all obtained from Uniprot; March 2015) and with results at 1% FDR. LFQ minimum ratio count was set to two and matching between runs was enabled with a window of 0.7 min. LFQ data were analyzed in Perseus[63] (version 1.5.3.0).

**Modeling and statistical analysis**. Subset selection for the linear multivariable models was done with the mlr package[64] in R using an exhaustive search over all sub-models and evaluating each sub-model by minimizing the mean squared error (MSE) on the test data through 100 times repeated cross-validation (tenfold, ten repeats)[65]. Model performance was reported as measured by the coefficient of determination ($R^2 = 1 - MSE/MST$, where MST is mean square total), evaluated on the hold-out dataset in cross-validation.

We also used several more complex and powerful machine-learning approaches that typically provide strong predictive performance[65]. These models were tuned using the same cross-validation approach and model performance assessment as for the linear model. Further details are provided in the Supplementary Information. For the linear model, we assessed the importance of each predictor remaining in the best-performing model using the PMVD method described in ref. [66] using the relaimpo package[67]. Statistical analysis on scatter plots was carried out using GraphPad Prism software. Differences were considered significant when p values were <0.05. Spearman rank correlation tests were performed in R using the cor and cor.test functions.

**Reporting summary**. Further information on research design is available in the Nature Research Reporting Summary linked to this article.

## Data availability
Immunopeptidomics and proteomics datasets analyzed in this study have been deposited to the ProteomeXchange Consortium via the PRIDE partner repository[68] with the dataset identifiers PXD012728 (MHC and SWATH data) and PXD012776 (LFQ data). MRM data have been made available on the The PeptideAtlas Project[69] at the following address: http://www.peptideatlas.org/PASS/PASS01317. Data underlying Fig. 5 and Supplementary Fig. 6 are provided as a Source Data File. Data underlying Figs. 1–4 and Supplementary Figs. 4 and 5 are included as Supplementary Data Files 1 and 2. All other data are available from the corresponding authors upon reasonable requests.

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

## Acknowledgements

We thank Stephen J. Turner and Jamie Rossjohn for critical reading of the manuscript, Lukasz Kedzierski and Claerwen Jones for technical assistance, and staff at Monash University flow cytometry, biomedical proteomics and animal core facilities. We thank Dr. Ken Rock for provision of DC2.4 cells and Dr. Acha-Orbea for provision of Mutu DCs. This work was supported by a Sylvia and Charles Viertel Senior Medical Research Fellowship, an Australian Research Council (ARC) Future Fellowship FT170100174, and a National Health and Medical Research Council (NHMRC) Program grant APP1071916 (to N.L.L.G.), an NHMRC Principal Research Fellowship APP1137739 (to A.W.P.), an NHMRC Senior Research Fellowship APP1104329 (to D.C.T.) and an NHMRC Project grant APP1084283 (to A.W.P., N.P.C., and D.C.T.). A.H. and P.G.T. were partially supported by NIH/NIAID grant U19AI117891.

## Author contributions

Conceptualization N.L.L.G., A.W.P., N.P.C.; Methodology J.G., L.M.W., N.L.L.G., A.W.P., N.P.C., D.C.T., A.H., P.G.T., A.S.; Investigation J.G., T.W., X.Y.X.S., J.S., A.H.; Resources D.C.T., A.H., L.M.W., J.S., A.S., A.W.P., N.P.C., N.L.L.G.; Writing—original draft N.P.C., N.L.L.G., A.W.P.; Writing—review and editing N.P.C., N.L.L.G., A.W.P., P.G.T., A.H., D.C.T., L.M.W., A.S., J.S., T.W., J.G., X.Y.X.S.; Supervision N.L.L.G., N.P.C., A.W.P.; Funding acquisition A.W.P., N.P.C., D.C.T., N.L.L.G.
