## [Peer Review File · Nature Communications]

Reviewers' comments:

Reviewer #1 (Remarks to the Author):

I have been asked to particularly comment on the modeling and statistical analyses (linear multi-variable models and boosted regression tree and support vector machine). If I understand your paper correctly variable selection is the focus rather than model fit. You achieve this with your linear multi-variable models, but not with your ksvm and gbm analyses. It would be interesting to establish the relative importance of predictors for these methods as well as for other methods such as a random forest, a LASSO regression etc.

You report that the gradient boosting and support vector machines produce inferior fit to the best linear multi-variable model, presumably on the basis of the cross-validated coefficient of determination. However, no results are presented. Also the coefficient of determination seems an unusual measure to be using in this context. It is more usual to consider a variety of maximum likelihood measures that include a penalty for over-parametrization, such as the Schwarz Bayesian Criterion.

Reviewer #2 (Remarks to the Author):

Comments on manuscript Nature Communications (NCOMMS-18-38056) by Wu et al. Direct quantification of viral epitope abundance reveals the importance of direct and cross-presentation in driving CTL responses during influenza infection.

Dear Editor,

In this manuscript, Wu et al. describe the identification of a number of new MHC epitopes that are derived from influenza A virus. The infected culture mouse and human cells with the virus and analyzed the presented MHC peptides. They also quantified the identified flu peptides, which are presented directly by the cells. In addition, they checked which of these peptides are cross-presented by other antigen presenting cells and the quantified the levels of the directly presented peptides and of the cross-presented peptides. Furthermore, the authors have immunized mice with the peptides and checked the T cell responses of the mice by following intracellular production of interferon and of TNF. They also checked the production kinetics of the MHC peptides and the expression kinetics of the viral proteins and compared between these two. The conclusions of the manuscript are that there is no correlation between the levels of expression of the source proteins

and their derived peptides, yet, some correlation was observed between the affinity of the peptides to the MHC and their immunogenicity.

In my opinion, this manuscript is not ready for publication in its current form, even though the data is interesting and important. In my opinion, there are a number of issues that need to be resolved before accepting it.

First, I do not think that Nature Communication should accept manuscript without their associated raw data and full supplementary data. The addition of the spectra in the supplementary material does not provide the reader with sufficient information to evaluate the correctness of the data. Please see the accepted instruction for publication of MHC peptide data in Lillet al. (2018) Minimal Information About an Immuno-Peptidomics Experiment (MIAIPE). *Proteomics* 18, e1800110.

A deposition in a public database and addition of the entire list of identified peptides with their quantification data will be sufficient.

The third issue is the problem of the kinetics. The authors present the change in the level of the peptides relative to the proteins in percentages and this is an incorrect calculation for the sake of understanding how many of these molecules are derived from DRIPS. There could be a 1000 protein molecules made in each cell and this would constitute 100% and would take time, while there could be just 10 molecules of a peptide derived from the protein and this could be 100% of the peptide level. To supply 10 protein for degradation is not a problem if a 1000 protein molecules are made during the same time, and the resulting peptides are not products of DRIPS in such a case. The current assumption in the field is that proteins are made to full size and some of them are later degraded, likely since these are not assembled into the protein complexes or into the mature viruses.

Another issue is the measurement of immunogenicity, some of the peptides induced an immune reaction in a one or a few of the mice. It would be a good idea to explain how comes a 'non-self' peptide does not induce an immune reaction in most mice.

Some of the flu peptides identified here are longer than the binding consensus of these alleles. I suggest improving the explanation, on the observation of longer peptides seen after virus infection, if there is an explanation for that.

In addition, it should be described how the authors know that the peptides that they detect are not contaminating peptides that are generated by degradation outside of the cells of the proteins of the infecting virus particles. It is very likely that some of the identified peptides are generated by simple contamination by sporadic degradation products of the viral protein and are not ligands of the MHC. For example, peptides with extended lengths at either end are commonly seen in MHC peptidome analyses and are likely just contaminating peptides. It is advisable in my opinion to address this issue.

Please re-check the papers cited in the manuscript, since some are not cited properly.

In conclusion, I think that the manuscript is important and contains interesting data and ideas and that it should be improved before acceptance for publication in Nature Comm.

Reviewer #3 (Remarks to the Author):

Wu et al. performed a comprehensive qualitative and quantitative study of influenza class I MHC epitopes directly and cross-presented in professional antigen presenting cells and infected cells in infection. Data collected by mass spectrometry to identify and quantitate epitope abundance and assess kinetics of presentation, by binding assay to determine epitope affinity for MHC I, and by flow cytometry to characterize T cell responses are integrated to extract information about antigen presentation variables that influence CTL response hierarchy. The data confirm and provide novel insights. For example, previously recognized CTL epitopes were found and their pattern of dominance was confirmed. At the same time, novel epitopes were discovered including peptides of canonical and non-canonical lengths. More importantly, and most strikingly, the authors report no correlation between epitope abundance and T cell response while confirming a correlation between peptide:MHC I and immunogenicity.

The study also goes further to challenge our understanding about how immunodominance is established with the finding of an immunodominant peptide that is (relatively) abundant in cross-presentation but poorly presented by direct presentation. The contribution of cross-presentation in comparison to direct presentation will require further investigation. This study opens up a path to begin to understand how distinct peptides are differentially presented. Overall, the work surveys several parameters of antigen presentation in an elegant quantitative system that makes it possible to draw several conclusions about their relative importance with confidence. The manuscript is very well written, logically structured and a pleasure to read.

While it can be said that eluting peptides from MHC molecules is not new (see "Trends in immunology" article by Chicz and Rotchke dating from the time that they were both in the Strominger lab), MS-based HLA peptidomics is definitely a huge advance for the immunology field -- especially with regard to the potential for definition of novel T cell epitopes as illustrated in this paper. It would be nice to see the authors mention caveats when describing their approach and its contributions to epitope identification. Otherwise the reader may get the impression that this 'shiny new' method is the most accurate when, in fact, a combination of approaches should probably be recommended instead.

Mass Spec Sequence Bias: The authors suggest that this approach is not as biased or circular as other approaches, however, the approach is not entirely free from any biases. Specifically, peptides that are too hydrophobic or too hydrophilic might be missed. We have found this to be true in particular for peptides that are well defined using other methods but not recovered in peptide elution studies.

Furthermore peptides with cysteines and those that have other features that make them incompatible with ionization or lead to poor fragmentation may be missed in the Mass Spec approach.

T cell repertoire shaping: Perhaps more important, the authors seem to indicate that immunogenicity as observed in mice is considered to be equivalent to immunogenicity that might be observed in humans. Here again, some hubris - the T cell repertoire is shaped by exposure to vaccination and infection. Immune responses in naive mice cannot be considered to be entirely reflective of expected immune responses in non-naïve humans. Again, a comment to that effect would be appreciated by this reviewer.

The following points will further enhance this already excellent report:

1. The unexpected but oft-observed complexity of immune responses in syngeneic mice is observed in high definition in this study, which prompts the authors to raise the need to “reflect” (page 25) on this complexity. The authors should use this opportunity to begin such a discussion. Discuss (and speculate on) the significance of the studies to understanding human immune responses and to understanding recall immune responses upon secondary exposure.
2. As the authors note, the lack of correlation between epitope abundance and peptide-MHC binding affinity shown in Figure 4 is surprising. It is certainly a notable finding but only scratches the surface. The authors should dig deeper and discuss, for example, the affinities and abundances of the immunodominant peptides and how together these factors may contribute to explain their immunodominance. Also, how the ability of a peptide to bind one versus two MHC I alleles influences the correlation and impacts CTL response.
3. Given the quantitative data collected in terms of both peptides presented and antigen production over time after infection, speculate about the percent of antigen that is processed for presentation, if possible.
4. The authors overreach when they claim they define “the full spectrum of MHCI-bound IAV-derived peptides presented following infection” (page 5). Proteasome-catalyzed spliced IAV peptides may be generated and were neither identified nor ruled out here. Revision to say, for example, the full spectrum of MHCI-bound contiguous IAV-derived peptides would be more accurate. In a similar

vein, “the full spectrum” does not acknowledge limitations of the mass spectrometry approach that could mask detection of presented peptides.

5. Yet another overreach is the claim that the seven novel epitopes “comprise ~10% of the total anti-viral response” (page 17) when spliced peptide-specific CD8 T cells are unaccounted. It is also unclear where the authors measure the full flu virus CD8 T cell response. Without that information, at most a statement can be made about the contribution the novel epitopes make relative to the cumulative response observed to the epitopes measured.

6. The inset graph in Figure 3c will be clearer with a label indicating the DC2.4 and LET1 data were collected in a direct presentation measurement.

7. Reference the studies that report differences between peptide repertoires generated by the immunoproteasome and standard proteasome (page 25).

Reviewer #1 (Remarks to the Author):

I have been asked to particularly comment on the modeling and statistical analyses (linear multi-variable models and boosted regression tree and support vector machine). If I understand your paper correctly variable selection is the focus rather than model fit. You achieve this with your linear multi-variable models, but not with your ksvm and gbm analyses. It would be interesting to establish the relative importance of predictors for these methods as well as for other methods such as a random forest, a LASSO regression etc.

We were interested in both modeling aspects, namely the overall predictive power of the models and to understand which predictors contributed most to our observed immunodominance hierarchy. The subset selection component of our analysis provided information regarding variable importance, the svm and gbm methods were used to understand maximum predictive power of the model. The latter models include all predictors. We can compute variable importance for those models. Indeed, we performed such computations. However, since none of those models were as good as the linear model, we decided that discussing the importance of specific variables for an inferior model that should not be used might be confusing. In the revised manuscript, we now state this reasoning explicitly. We also include results from a random forest model. A LASSO regression is another way of doing a linear regression with certain penalties. Thus, while it is not testing if a nonlinear model fits better, we decided to include it in this revision to compare differences between our subset selection approach and LASSO as suggested by the reviewer. The following changes have been made to the manuscript (underlined):-

“Finally, to determine if a model that allowed more complex relations (i.e. beyond linear) between CTL magnitude and epitope-specific variables would perform better, we applied a support vector machine model (SVM), a random forest (RF), and a gradient boosted regression tree model (GBM). We also used a LASSO method as an alternative way of doing variable selection for a linear model. However, none of these models provided better performance than the multivariate linear model (measured by cross-validated R^2) after tuning and training. Further details are provided in the Supplementary Information. Thus, for the dataset analysed here, cross-presentation and the IC_{50} variables additively determine T cell response strength, with no further predictive strength gained from the other variables or from a more complicated model structure.”

You report that the gradient boosting and support vector machines produce inferior fit to the best linear multi-variable model, presumably on the basis of the cross-validated coefficient of determination. However, no results are presented.

We now include details regarding the performance of those methods in Supplementary Information.

Also the coefficient of determination seems an unusual measure to be using in this context. It is more usual to consider a variety of maximum likelihood measures that include a penalty for over-parametrization, such as the Schwarz Bayesian Criterion.

We want to emphasize that our performance measure is always the *cross-validated* coefficient of determination, i.e. that quantity determined for repeated hold-out/test sets. We agree that model evaluation using the coefficient of determination applied to the data that was used for fitting would be unusual and misleading. The approach suggested by the reviewer of evaluating

model performance on the data that was used for fitting/training using a likelihood measure, and adding a penalty for over-parameterization, is a valid alternative. However, in our experience the direct evaluation of predictive performance through cross-validation tends to produce more reliable estimates of actual model performance. We thus decided to use this approach. Because of this performance evaluation through cross-validation, there is no need to use one of the common measures (AIC, BIC, etc.) that try to penalize/reduce overfitting.

Reviewer #2 (Remarks to the Author):

I do not think that Nature Communication should accept manuscript without their associated raw data and full supplementary data. The addition of the spectra in the supplementary material does not provide the reader with sufficient information to evaluate the correctness of the data. Please see the accepted instruction for publication of MHC peptide data in Lillet al. (2018) Minimal Information About an Immuno-Peptidomics Experiment (MIAIPE). Proteomics 18, e1800110.

A deposition in a public database and addition of the entire list of identified peptides with their quantification data will be sufficient.

We apologise that this was not clear in our submitted version of the manuscript. As co-authors of the MIAIPE recommendations we of course agree and have now made this explicit in the manuscript, and we have provided two supplementary datasets with additional information on peptide identification and quantification (Supplementary Data 1 and 2). In addition, we have now deposited all raw and searched mass spectrometry data into the PRIDE and Peptide Atlas repositories, as now detailed in a dedicated section of the paper.

“Data Availability

Immunopeptidomics and proteomics datasets analysed in this study have been deposited to the ProteomeXchange Consortium via the PRIDE partner repository¹¹³ with the data set identifiers PXD012728 (MHC and SWATH data, accessible via username reviewer54443@ebi.ac.uk and password hyffczAI) and PXD012776 (LFQ data, accessible via username reviewer81797@ebi.ac.uk and password XPemWOCJ). MRM data have been made available on the The PeptideAtlas Project¹¹⁴ at the following address: <http://www.peptideatlas.org/PASS/PASS01317>.”

The third issue is the problem of the kinetics. The authors present the change in the level of the peptides relative to the proteins in percentages and this is an incorrect calculation for the sake of understanding how many of these molecules are derived from DRIPS. There could be a 1000 protein molecules made in each cell and this would constitute 100% and would take time, while there could be just 10 molecules of a peptide derived from the protein and this could be 100% of the peptide level. To supply 10 protein for degradation is not a problem if a 1000 protein molecules are made during the same time, and the resulting peptides are not products of DRIPS in such a case. The current assumption in the field is that proteins are made to full size and some of them are later degraded, likely since these are not assembled into the protein complexes or into the mature viruses.

While we did determine absolute peptide levels, it was not feasible to determine the precise amount of protein degraded to liberate detected epitopes using the methods applied in our study. We have included a detailed outline of relative protein quantitation in the revised manuscript. The intention of this figure was not to determine the efficiency of peptide

generation from source proteins, but to convey the relative kinetics of each, to provide an indication of the source of the peptides (i.e. mature proteins or DRiPs/newly synthesized protein). We respectfully contend that our observation that the peak peptide abundance often precedes peak protein abundance suggests that a significant proportion of the peptide is derived from DRiPs or rapidly degraded protein, as stated in the current manuscript.

Another issue is the measurement of immunogenicity, some of the peptides induced an immune reaction in a one or a few of the mice. It would be a good idea to explain how comes a ‘non-self’ peptide does not induce an immune reaction in most mice.

The underlined section has been included in the Discussion:-

“Secondly, our mass spectrometry based approach identified a number of new IAV-specificities that all were immunogenic in at least one mouse, and that for many subdominant epitopes there was a private response where recruitment of these CTL specificities into the immune repertoire appears to be more stochastic than the responses to the more dominant T cell determinants. The most plausible explanation for poor or sporadic responses despite detectable epitope presentation is that these represent epitopes for which there are no, or exceedingly few, naïve T cells available. So-called ‘holes in the T cell repertoire’ have been attributed, in part, to the deletion of cross-reactive T cells due to excessive similarity to self-peptides^{67, 68}. Alternatively, there are epitopes that elicit poor T cell responses despite the availability of a large number of specific T cells being present in the naïve repertoire. In this case, the subdominant nature of the response may be a consequence of the poor ‘quality’ of the available repertoire, due either to a low affinity TCR-pMHC interaction³⁹ or a non-canonical TCR-pMHC interaction that is incapable of driving robust signaling⁶⁹.”

Some of the flu peptides identified here are longer than the binding consensus of these alleles. I suggest improving the explanation, on the observation of longer peptides seen after virus infection, if there is an explanation for that.

We have included an extended discussion of the longer peptides, as follows:-

“Of the 21 peptides identified in this study, seven were 10-11aa in length, including four of the seven novel peptides. Notably, all of these longer IAV-derived peptides were bound by H-2D^b while all seven of the shorter octamers were bound by H-2K^b, consistent with previous observations that MHC alleles exhibit distinct peptide length preferences^{46, 47}. The identification of 10-11aa peptides is also consistent with the observation that a substantial proportion of naturally occurring mouse MHCI-bound peptides are longer than the canonical length of 8-9aa. Such observations are becoming more apparent as global immunopeptidomics analyses increase⁴⁸⁻⁵⁰, and underscore the potential deficiencies in many predictive algorithms^{10, 51-54}. That long peptides from viruses and tumors are naturally processed and presented on MHCI, suggests they likely play a key role in antiviral and anti-tumor immunity⁵⁵⁻⁵⁷. Long peptides have been found to be accommodated in the MHCI predominantly via a central bulging from the antigen binding cleft^{57, 58} or occasionally via amino- or carboxy-terminal extensions from the MHCI groove^{49, 59, 60}. Structural analysis of TCRs binding long peptides in complex with MHCI show that TCRs can achieve pMHC recognition using a variety of strategies, including ‘flattening’ the bulged peptide to increase MHCI contact, or sitting atop the peptide making minimal MHCI contact^{57, 58}. Collectively, given their prevalence and the demonstrated ability of T cells to recognize long peptides, our data further

suggest that proteome-wide discovery-based approaches should now be utilized to obtain a comprehensive representation of the peptidome.”

In addition, it should be described how the authors know that the peptides that they detect are not contaminating peptides that are generated by degradation outside of the cells of the proteins of the infecting virus particles. It is very likely that some of the identified peptides are generated by simple contamination by sporadic degradation products of the viral protein and are not ligands of the MHC. For example, peptides with extended lengths at either end are commonly seen in MHC peptidome analyses and are likely just contaminating peptides. It is advisable in my opinion to address this issue.

We are confident the peptides detected are not contaminants from the cellular antigens. Firstly, the K^b and D^b complexes were individually immunoprecipitated with allomorph specific monoclonal antibodies and the captured pMHC washed stringently before release of the bound peptides. Secondly the peptides adhere to allele specific binding motifs. Thirdly, and perhaps the most compelling observation resides in the distinct peptidomes we observed for K^b and D^b – if there were contaminants these would be present in both IPs and would be easily detected using the highly sensitive MRM assays. Indeed, we only observed one incidence where a peptide was detected in both K^b and D^b immunoprecipitates derived from HA₄₁₋₄₉. The propensity of this peptide to bind to both alleles is corroborated by its promiscuous binding in affinity measurement analysis.

Please re-check the papers cited in the manuscript, since some are not cited properly.

This has been checked and rectified.

In conclusion, I think that the manuscript is important and contains interesting data and ideas and that it should be improved before acceptance for publication in Nature Comm.

Reviewer #3 (Remarks to the Author):

Wu et al. performed a comprehensive qualitative and quantitative study of influenza class I MHC epitopes directly and cross-presented in professional antigen presenting cells and infected cells in infection. Data collected by mass spectrometry to identify and quantitate epitope abundance and assess kinetics of presentation, by binding assay to determine epitope affinity for MHC I, and by flow cytometry to characterize T cell responses are integrated to extract information about antigen presentation variables that influence CTL response hierarchy. The data confirm and provide novel insights. For example, previously recognized CTL epitopes were found and their pattern of dominance was confirmed. At the same time, novel epitopes were discovered including peptides of canonical and non-canonical lengths. More importantly, and most strikingly, the authors report no correlation between epitope abundance and T cell response while confirming a correlation between peptide:MHC I and immunogenicity.

The study also goes further to challenge our understanding about how immunodominance is established with the finding of an immunodominant peptide that is (relatively) abundant in cross-presentation but poorly presented by direct presentation. The contribution of cross-presentation in comparison to direct presentation will require further investigation. This study opens up a path to begin to understand how distinct peptides are differentially presented. Overall, the work surveys several parameters of antigen presentation in an elegant quantitative

system that makes it possible to draw several conclusions about their relative importance with confidence. The manuscript is very well written, logically structured and a pleasure to read.

While it can be said that eluting peptides from MHC molecules is not new (see "Trends in immunology" article by Chicz and Rotchke dating from the time that they were both in the Strominger lab), MS-based HLA peptidomics is definitely a huge advance for the immunology field -- especially with regard to the potential for definition of novel T cell epitopes as illustrated in this paper. It would be nice to see the authors mention caveats when describing their approach and its contributions to epitope identification. Otherwise the reader may get the impression that this 'shiny new' method is the most accurate when, in fact, a combination of approaches should probably be recommended instead.

We absolutely agree that MS-based HLA peptidomics is not new, but would argue that the technology has now matured to allow more biologically driven hypotheses to be addressed. Indeed, combining immunopeptidomics for pMHC purification and sequencing with extensive characterisation of peptide immunogenicity is an area that has seen little intensive investigation to date (*e.g.* see Croft et al, 2019, PNAS; PMID:30718433). Further, we believe that using targeted MS to dissect changes in peptide abundance/diversity between direct and cross-presentation clearly remains a novel aspect of our study. However, the reviewer is correct that the methods used lend themselves to some degree of bias, and we have reflected upon that in the following statement in the Discussion:

“Whilst we have conceivably captured all available H-2D^b and K^b complexes from cells through immunoprecipitation, there is likely to be some bias in peptide identification. For example, C₁₈ chromatography may miss overly hydrophilic or hydrophobic sequences and due to the stochastic nature of conventional MS modalities for peptide isolation and fragmentation - coupled to protein-centric search algorithms - we may fail to detect and assign peptides of lower abundance, of poor ionization, or where ambiguous spectra have been acquired⁴⁴. It is also conceivable that a fraction of the viral repertoire is presented through proteasome-catalysed peptide splicing, an area that will require future investigation now that accessible algorithms have been made available⁴⁵.”

Mass Spec Sequence Bias: The authors suggest that this approach is not as biased or circular as other approaches, however, the approach is not entirely free from any biases. Specifically, peptides that are too hydrophobic or too hydrophilic might be missed. We have found this to be true in particular for peptides that are well defined using other methods but not recovered in peptide elution studies. Furthermore peptides with cysteines and those that have other features that make them incompatible with ionization or lead to poor fragmentation may be missed in the Mass Spec approach.

Please see response to previous comments.

T cell repertoire shaping: Perhaps more important, the authors seem to indicate that immunogenicity as observed in mice is considered to be equivalent to immunogenicity that might be observed in humans. Here again, some hubris - the T cell repertoire is shaped by exposure to vaccination and infection. Immune responses in naive mice cannot be considered to be entirely reflective of expected immune responses in non-naïve humans. Again, a comment to that effect would be appreciated by this reviewer.

We have now included the following paragraph in the Discussion.

“The relevance of this study to the immune CD8⁺ T cell responses in humans should be highlighted, especially in light of the fact that many human responses to pathogens, and IAV in particular, are not primary immune responses, but rather reflect the recall response of memory CD8⁺ T cells that have been exposed to IAV multiple times over a life course. Nonetheless, humans must acquire T cell immunity to viruses, and the primary CTL response to the initial exposure event both defines the ability to efficiently clear the acute primary infection, and establishes a framework from which an individual’s capacity to respond to virus rechallenge thereafter is defined. At a simplistic level, the primary response will determine whether subsequent responses are derived from naïve or memory populations⁹⁰. Moreover, studies suggest that repeated exposures to the same virus tend to exacerbate or focus responses that were established following primary infection^{91, 92}. Finally, original antigenic sin refers to the observation that memory established upon primary exposure can detrimentally impact on subsequent responses, and has been observed in antiviral CD8⁺ T cell responses^{93, 94}. Collectively, an understanding of the key contributors to antigenicity and immunodominance in primary CD8⁺ T cell responses is imperative for a comprehensive understanding of both primary and recall CD8⁺ T cell responses in humans”

The following points will further enhance this already excellent report:

1. The unexpected but oft-observed complexity of immune responses in syngeneic mice is observed in high definition in this study, which prompts the authors to raise the need to “reflect” (page 25) on this complexity. The authors should use this opportunity to begin such a discussion. Discuss (and speculate on) the significance of the studies to understanding human immune responses and to understanding recall immune responses upon secondary exposure.

Please see response above.

2. As the authors note, the lack of correlation between epitope abundance and peptide-MHC binding affinity shown in Figure 4 is surprising. It is certainly a notable finding but only scratches the surface. The authors should dig deeper and discuss, for example, the affinities and abundances of the immunodominant peptides and how together these factors may contribute to explain their immunodominance. Also, how the ability of a peptide to bind one versus two MHC I alleles influences the correlation and impacts CTL response.

We have added the below paragraph into the discussion prior to an existing discussion of how peptide affinity for MHC may influence CTL responses independently of epitope abundance. Please note that we were unable to discuss how the relative affinities and abundances of the HA₄₁₋₄₉ peptide influenced response magnitude as, using peptide restimulation to induce cytokine production, we could not discern the proportion of the response driven by the D^b-bound vs K^b-bound peptide.

“Surprisingly, and in accordance with our recent findings for VACV⁸⁵, our data revealed that the strong correlation between peptide affinity for MHCI (as measured by IC₅₀) and T cell response magnitude occurred independently of epitope abundance. This is exemplified by the relative levels of the HA₄₁₋₄₉ peptide, which bound both H2D^b and K^b, but exhibited a substantially greater affinity for K^b. In spite of this, the relative abundance of presented peptide

derived from each of the MHC alleles was similar, following both direct and cross-presentation. Additionally, the affinities of the NP₃₆₆ and PA₂₂₄ peptides for H2D^b were the highest observed, corresponding with their immunodominant response status. While this was also reflected by abundant NP₃₆₆ presentation, PA₂₂₄ was amongst the least abundantly presented peptides following direct presentation.”

3. Given the quantitative data collected in terms of both peptides presented and antigen production over time after infection, speculate about the percent of antigen that is processed for presentation, if possible.

Please see response to Reviewer 2, point 2.

4. The authors overreach when they claim they define “the full spectrum of MHCI-bound IAV-derived peptides presented following infection” (page 5). Proteasome-catalyzed spliced IAV peptides may be generated and were neither identified nor ruled out here. Revision to say, for example, the full spectrum of MHCI-bound contiguous IAV-derived peptides would be more accurate. In a similar vein, “the full spectrum” does not acknowledge limitations of the mass spectrometry approach that could mask detection of presented peptides.

We have made the following modification at the beginning of the Results section:-

“To define a more complete spectrum of MHCI-bound IAV-derived peptides presented following infection,…”

We have also added a paragraph in the discussion to highlight potential bias in this approach (please see response to Reviewer’s earlier comment above).

5. Yet another overreach is the claim that the seven novel epitopes “comprise ~10% of the total anti-viral response” (page 17) when spliced peptide-specific CD8 T cells are unaccounted. It is also unclear where the authors measure the full flu virus CD8 T cell response. Without that information, at most a statement can be made about the contribution the novel epitopes make relative to the cumulative response observed to the epitopes measured.

We have amended the sentence as follows:-

“Collectively, CD8⁺ T cell responses to the seven novel epitopes identified in this study comprise ~10% of the total anti-viral response, as mapped thus far.”

6. The inset graph in Figure 3c will be clearer with a label indicating the DC2.4 and LET1 data were collected in a direct presentation measurement.

We have amended the graph to indicate this.

7. Reference the studies that report differences between peptide repertoires generated by the immunoproteasome and standard proteasome (page 25).

The following references have been added:-

J Driscoll, MG Brown, D Finley, JJ Monaco. MHC-linked LMP gene products specifically alter peptidase activities of the proteasome *Nature* 1993. **365**:262–264.

Gaczynska, M., Rock, K. L., Spies, T. and Goldberg, A. L. Peptidase activities of proteasomes are differentially regulated by the major histocompatibility complex-encoded genes for LMP2 and LMP7. *Proc. Natl. Acad. Sci. USA* 1994. **91**: 9213-9217

RE Toes, AK Nussbaum, S Degermann, M Schirle, NP Emmerich, M Kraft, C Laplace, A Zwinderman, TP Dick, J Müller, B Schönfish, C Schmid, HJ Fehling, S Stevanovic, HG Rammensee, H Schild. Discrete cleavage motifs of constitutive and immunoproteasomes revealed by quantitative analysis of cleavage products. *J Exp Med.* 2001. **194**:1-12

Reviewers' comments:

Reviewer #1 (Remarks to the Author):

Thank you. You have addressed all my concerns to my satisfaction.

Reviewer #2 (Remarks to the Author):

Comments on NCOMM-18-38056A by Dr. La Gruta et al.

In general, the authors have responded well, in my opinion, to most of the reviewers' comments.

The two issues that are not fully addressed are:

1. The issue of the analysis of the MHC peptides as derived from defective ribosome products proteins or rapidly degrading proteins. The authors have shown that the rate of reaching 100% of the levels of presentation of the MHC is often faster than the rate of reaching 100% of the cellular levels of the viral proteins. They plot the two rates on the same graph as percentages (reaching 100%), but this graph is confusing in my opinion, since there are just a few presented MHC peptides relative to the numbers of viral proteins produced by the same cells. Therefore, 100% of the proteins and of the MHC peptides are vastly different. Maybe, the authors can add a graph with the numbers of proteins per cell, and MHC peptides per cell, using a log scale so that the two vastly different numbers will fit one graph. In my opinion, the statement on page 28 "supporting the notion that MHCI binding peptides are predominantly sourced from defective proteins or pioneer translation products, rather than mature stable proteins" is not supported by the data in the manuscript (in Figure 2).
2. The comment that the immunogenicity test of the peptides in a mouse system is not a good indication for the process of generating immune reaction and memory in a human IAV infection was corrected to some extent in this version of the manuscript. It would be nice in my opinion if the authors add a suggestion on how to test the process of generation of immunity to this virus in humans, after their added text in the Discussion: "Collectively, an understanding of the key contributors to antigenicity and immunodominance in primary CD8+ T cell responses is imperative for a comprehensive understanding of both primary and recall CD8+ T cell responses in humans."

Reviewer #3 (Remarks to the Author):

The authors adequately addressed all the points raised in this review and the others. I recommend the manuscript to be published.

Reviewer #2 (Remarks to the Author):

In general, the authors have responded well, in my opinion, to most of the reviewers' comments.

The two issues that are not fully addressed are:

1. The issue of the analysis of the MHC peptides as derived from defective ribosome products proteins or rapidly degrading proteins. The authors have shown that the rate of reaching 100% of the levels of presentation of the MHC is often faster than the rate of reaching 100% of the cellular levels of the viral proteins. They plot the two rates on the same graph as percentages (reaching 100%), but this graph is confusing in my opinion, since there are just a few presented MHC peptides relative to the numbers of viral proteins produced by the same cells. Therefore, 100% of the proteins and of the MHC peptides are vastly different. Maybe, the authors can add a graph with the numbers of proteins per cell, and MHC peptides per cell, using a log scale so that the two vastly different numbers will fit one graph. In my opinion, the statement on page 28 "supporting the notion that MHC I binding peptides are predominantly sourced from defective proteins or pioneer translation products, rather than mature stable proteins" is not supported by the data in the manuscript (in Figure 2).

As we stated in our earlier response, our protein analysis for the infection timecourse was by SWATH-MS, a form of label-free analysis that allowed only for quantitation of relative, and not absolute, protein abundance. Therefore, a direct comparison of peptide vs protein abundance is not possible and we cannot provide the requested graphs. We would like to reiterate that the key point we are attempting to convey with these data is not reliant upon absolute values; rather, we are tracking the rate of detection of peptide-MHC and viral protein synthesis. Relating these two variables for virus infection has been established previously [Croft PLoS Pathog 2013].

With regard to the statement referred to here, i.e. "supporting the notion that MHC I binding peptides are predominantly sourced from defective proteins or pioneer translation products, rather than mature stable proteins"; while we agree with the reviewer that this observation is by no means a definitive demonstration that presented peptides are predominantly sourced from DRiPs, we believe the fact that peak antigen expression does not predict peak epitope presentation is an important observation that deserves comment. For this reason we have altered our wording to (underlined) :-

"For later timepoints, (≥ 3 hpi) the peak presentation of all peptides either preceded or was coincident with peak protein expression, suggesting that antigen presentation is linked with the translation of newly synthesised polypeptides. This would be consistent with any mechanism that allows for the sampling of newly translated proteins, including the disposal of defective products of translation or those that fail to achieve their final function in the viral lifecycle⁷⁹."

2. The comment that the immunogenicity test of the peptides in a mouse system is not a good indication for the process of generating immune reaction and memory in a human IAV infection was corrected to some extent in this version of the manuscript. It would be nice in my opinion if the authors add a suggestion on how to

test the process of generation of immunity to this virus in humans, after their added text in the Discussion: “Collectively, an understanding of the key contributors to antigenicity and immunodominance in primary CD8+ T cell responses is imperative for a comprehensive understanding of both primary and recall CD8+ T cell responses in humans.”

We have added the following statement to the end of this paragraph (underlined):-

“The relevance of this study to the immune CD8+ T cell responses in humans should be highlighted, especially in light of the fact that many human responses to pathogens, and IAV in particular, are not primary immune responses, but rather reflect the recall response of memory CD8⁺ T cells that have been exposed to IAV multiple times over a life course. Nonetheless, humans must acquire T cell immunity to viruses, and the primary CTL response to the initial exposure event both defines the ability to efficiently clear the acute primary infection, and establishes a framework from which an individual’s capacity to respond to virus rechallenge thereafter is defined. At a simplistic level, the primary response will determine whether subsequent responses are derived from naïve or memory populations⁸⁷. Moreover, studies suggest that repeated exposures to the same virus tend to exacerbate or focus responses that were established following primary infection^{88, 89}. Finally, original antigenic sin refers to the observation that memory established upon primary exposure can detrimentally impact on subsequent responses, and has been observed in antiviral CD8⁺ T cell responses^{90, 91}. Collectively, an understanding of the key contributors to antigenicity and immunodominance in primary CD8⁺ T cell responses is imperative for a comprehensive understanding of both primary and recall CD8⁺ T cell responses in humans. Certainly, characteristics of mouse and human antiviral CD8+ T cell responses (reproducible immunogenicity and immunodominance hierarchies between MHC matched individuals) are similar, and direct analyses of epitope presentation are becoming highly feasible in human cells, as evidenced by our recent findings⁹². Such studies will ultimately elucidate whether similar mechanisms drive CTL immunodominance in the human setting.”